# Towards Faithful Neural Network Intrinsic Interpretation with Shapley Additive Self-Attribution

## Abstract

Self-interpreting neural networks have garnered significant interest in research. Existing works in this domain often (1) lack a solid theoretical foundation ensuring genuine interpretability or (2) compromise model expressiveness. In response, we formulate a generic Additive Self-Attribution (ASA) framework. Observing the absence of Shapley value in Additive Self-Attribution, we propose Shapley Additive Self-Attributing Neural Network (SASANet), with theoretical guarantees for the self-attribution value equal to the output's Shapley values. Specifically, SASANet uses a marginal contribution-based sequential schema and internal distillation-based training strategies to model meaningful outputs for any number of features, resulting in un-approximated meaningful value function. Our experimental results indicate SASANet surpasses existing self-attributing models in performance and rivals black-box models. Moreover, SASANet is shown more precise and efficient than post-hoc methods in interpreting its own predictions. [1]

## 1 Introduction

While neural networks excel in fitting complex real-world problems due to their vast hypothesis space, their lack of interpretability poses challenges for real-world decision-making. Although post-hoc interpretation algorithms (Lundberg & Lee (2017); Shrikumar et al. (2017)) offer extrinsic model-agnostic interpretation, the un-transparent intrinsic modeling procedure unavoidably lead to inaccurate interpretation (Laugel et al. (2019); Frye et al. (2020)). Thus, there's a growing need for self-interpreting neural structures that intrinsically convey their prediction logic faithfully.

Self-interpreting neural networks research has garnered notable interest, with the goal of inherently elucidating a model's predictive logic. Various approaches are driven by diverse scenarios. For example, Alvarez Melis & Jaakkola (2018) linearly correlates outputs with features and consistent coefficients, while Agarwal et al. (2020) employs multiple networks each focusing on a single feature, and Wang & Wang (2021) classifies by comparing inputs with transformation-equivariant prototypes. However, many existing models, despite intuitive designs, might not possess a solid theoretical foundation to ensure genuine interpretability. The effectiveness of attention-like weights, for instance, is debated (Serrano & Smith (2019); Wiegreffe & Pinter (2019)). Additionally, striving for higher interpretability often means resorting to simpler structures or intricate regularization, potentially compromising prediction accuracy.

This paper aims to achieve theoretically guaranteed faithful self-interpretation while retaining expressiveness in prediction. To achieve this goal, we formulate a generic Additive Self-Attribution (ASA) framework. ASA offers an intuitive understanding, where the contributions of different observations are linearly combined for the final prediction. Notably, even with varying interpretative angles, many existing methods implicitly employ such structure. Thus, we utilize ASA to encapsulate and distinguish their interpretations, clarifying when certain methods are favored over others.

Upon examining studies through the ASA lens, we identified an oversight regarding the Shapley value. Widely recognized for post-hoc attribution (Lundberg & Lee (2017); Bento et al. (2021)) with robust theoretical backing from coalition game theory (Shapley et al. (1953)), its potential has

---

[1]Code: https://anonymous.4open.science/r/SASANet-B343

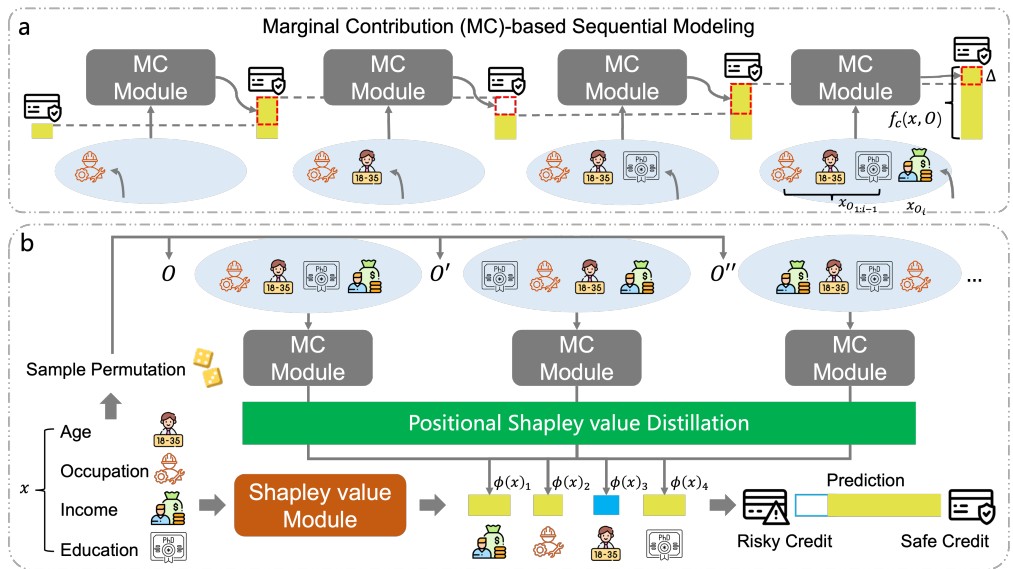

Figure 1: A schematic of the SASANet procedure. (a) Each sample is viewed as a set of features. The intermediate sequential module models an permutation-variant intermediate output as the cumulative contributions of these features. (b) The Shapley Value module trains a self-attributing network via internal Shapley value distillation, in which the attribution values proven to converge to the permutation-invariant final output's Shapley value.

been underutilized. Notably, while a recent work called Shapley Explanation Network (SHAPNet) (Wang et al. (2021)) achieves layer-wise Shapley attribution, it lacks model-wise feature attribution. Addressing this gap, we introduce a Shapley Additive Self-Attributing Neural Network (SASANet) under the ASA framework, depicted in Figure 1. In particular, we directly define value function as model output given arbitrary number of input features. With an intermediate sequential modeling and distillation-based learning strategy, SASANet can be theoretically proven to provide additive self-attribution converging to the Shapley value of its own output. Our evaluations of SASANet on various datasets show it surpasses current self-attributing models and reaches comparable performance as black-box models. Furthermore, compared to post-hoc approaches, SASANet's self-attribution offers more precise and streamlined interpretation of its predictions.

## 2 PRELIMINARIES

### 2.1 ADDITIVE SELF-ATTRIBUTING MODELS

Oriented for various tasks, self-interpreting networks often possess distinct designs, obscuring their interrelations and defining characteristics. Observing a commonly shared principle of linear feature contribution aggregation across numerous studies, we introduce a unified framework:

**Definition 2.1 (Additive Self-Attributing Model)** *Additive Self-Attributing Models output the sum of intrinsic feature attribution values that hold desired properties, formulated as*

$$f(x; \theta, \phi_0) = \phi_0 + \sum_{i \in \mathcal{N}} \phi(x; \theta)_i, \qquad s.t. \quad \mathcal{C}(\phi, x), \tag{1}$$

*where $x = [x_1, \cdots, x_N] \in \mathbb{R}^N$ denotes a sample with $N$ features, $\mathcal{N} = \{1, 2, \cdots, N\}$ denotes feature indices, $\phi(x; \theta)_i$ is the $i$-th feature's attribution value, $\phi_0$ is a sample-independent bias, $\mathcal{C}(\phi, x)$ denotes constraints on attribution values.*

As there are abundant ways to build additive models, attribution terms are required to satisfy specific constraints for ensuring the physically meaningfulness, often achieved through regularizers or structural inductive bias. Neglecting various pre- and post-transformations for adapting to specific inputs and outputs, many existing structures providing intrinsic feature attribution matches ASA framework. Below, we introduce several representative works.

**Self-Explaining Neural Network (SENN)** (Alvarez Melis & Jaakkola (2018)). SENN models generalized coefficients for different concepts (features) and constrains the coefficients to be

locally bounded by the feature transformation function. Using $x$ to denote the explained feature unit, SENN's attribution can be formulated as $f(x) = \sum_{i=1}^{N} a(x;\theta)_i x_i + b$, constraining that $a(x;\theta)_i$ is approximately the gradient of $f(x)$ with respect to each $x_i$. By setting $\phi(x;\theta) := [a(x;\theta)_1 x_1, \cdots, a(x;\theta)_N x_N] \in \mathbb{R}^N$, SENN aims at the form of Eq. 1 with a constraint $\nabla_{x_i} f(x;\theta,\phi_0) \approx \phi(x;\theta)_i / x_i$ for $i \in \mathcal{N}$. To realize the constraints, SENN adds a regularizer on the distance between the coefficients and the gradients.

**Neural Additive Model (NAM)** (Agarwal et al. (2020)). NAM models each feature's independent contribution and sums them for the prediction as $f(x) = \sum_{i=1}^{N} h_i(x_i;\theta_i) + b$. Regarding $h_i(\cdot;\theta_i) : \mathbb{R} \to \mathbb{R}$ as a function $\phi(\cdot;\theta)_i : \mathbb{R}^N \to \mathbb{R}$ constrained depend only on $x_i$, NAM targets the form of Eq. 1 with a constraint: $\forall x, x' \quad x_i = x'_i \to \phi(x;\theta)_i = \phi(x';\theta)_i$ for $i \in \mathcal{N}$. To realize such constraints, NAM models each feature's attribution with an independent network module.

**Self-Interpretable model with Transformation Equivariant Interpretation (SITE)** (Wang & Wang (2021)). SITE models the output as a similarity between the sample and a relevant prototype $G(x;\theta) \in \mathbb{R}^N$, formulated as $f(x) = \sum_{i=1}^{N} G(x;\theta)_i x_i + b$. Each feature's contribution term $G(x;\theta)_i x_i$ is the attribution value. The prototype is constrained to (1) resemble real samples of the corresponding class and (2) be transformation equivariant for image inputs. Thus, SITE targets the form of Eq. 1 with constraints: $\phi(x;\theta)_i = G(x;\theta)_i x_i$ for $i \in \mathcal{N}$, $G(x;\theta) \sim \mathcal{D}_c$, and $T_\beta^{-1}(G(T_\beta(x);\theta)) \sim \mathcal{D}_c$, where $\mathcal{D}_c$ is sample distribution, $T_\beta$ is predefined transformations. To realize such constraints, SITE regularizes sample-prototype distance and $T_\beta$'s reconstruction error.

**Salary-Skill Value Composition Network (SSCN)**(Sun et al. (2021)). Domain-specific studies also seek self-attributing networks. For example, SSCN models job salary as the weighted average of skills' values, formulated as $f(x;\theta_d,\theta_v) = \sum_{i=1}^{N} d(x;\theta_d)_i v(x_i;\theta_v)$, where $d(x;\theta_d)_i$ denotes skill dominance and $v(x_i;\theta_v)$ denotes skill value, defined as task-specific components in their study. Regarding each skill as a salary prediction feature, SSCN inherently targets the form of Eq. 1 with constraints: $\phi(x;\theta)_i = d(x;\theta_d)_i v(x;\theta_v)_i$, $\forall x, x' \quad x_i = x'_i \to v(x;\theta_v)_i = v(x';\theta_v)_i$, and $\sum_{i=1}^{N} d(x;\theta_d)_i = 1$ for $i \in \mathcal{N}$. To realize such constraints, SSCN models $v$ with a single-skill network and $d$ with a self-attention mechanism.

## 2.2 Shapley Additive Self-Attributing Model

In coalition game theory, Shapley value ensures *Efficiency*, *Linearity*, *Nullity*, and *Symmetry* axioms for fair contribution allocation, largely promoting post-hoc interpretation methods (Lundberg & Lee (2017); Bento et al. (2021)). However, there's a gap in studying self-attributing networks satisfying these axioms. Therefore, we define the Shapley ASA model as:

**Definition 2.2 (Shapely Additive Self-Attributing Model)** *Shapely Additive Self-Attributing Model are ASA models that are formulated as*

$$
\begin{aligned}
f(x;\theta,\phi_0) =& \phi_0 + \sum_{i=1}^{N} \phi(x;\theta)_i, \\
s.t. \quad \phi(x;\theta)_i =& \frac{1}{N!} \sum_{O \in \pi(\mathcal{N}), k \in \mathcal{N}} \mathbb{I}\{O_k = i\}(v_f(x_{O_{1:k-1} \cup \{i\}}; \theta, \phi_0) - v_f(x_{O_{1:k-1}}; \theta, \phi_0)),
\end{aligned}
\tag{2}
$$

*where $\pi(\cdot)$ denotes all possible permutations, $x_\mathcal{S} := \{x_i | i \in \mathcal{S}\}$ represents a subset of features in $x$ given $\mathcal{S} \in \mathcal{N}$ ($x = x_\mathcal{N}$), $v_f$ is a predefined value function about the effect of $x_\mathcal{S}$ with respect to $f$.*

Following coalition game theory (Shapley et al. (1953)), we constrain $\phi(x;\phi)_i$ to feature $x_i$'s average marginal contribution upon joining a random input subset, leading to Shapley value axioms.

**Theorem 2.3** *In a Shapley ASA model, the following axioms hold:*

**(Efficiency):** $\sum_{i \in \mathcal{N}} \phi(x;\theta)_i = v_f(x_\mathcal{N}; \theta, \phi_0) - v_f(x_\emptyset; \theta, \phi_0)$.

**(Linearity):** *Given Shapley ASA models $f$, $f'$, and $f''$, for all $\alpha, \beta \in \mathbb{R}$: if $v_{f''} = \alpha v_f + \beta v_{f'}$, then $\phi^{(f'')} = \alpha \phi^{(f)} + \beta \phi^{(f')}$, where $\phi^{(\cdot)}$ denotes the internal attribution value of a Shapley ASA model.*

**(Nullity):** *If $\forall \mathcal{S} \subset \mathcal{N} \backslash \{i\} \quad v_f(x_{\mathcal{S} \cup \{i\}}; \theta, \phi_0) = v_f(x_\mathcal{S}; \theta, \phi_0)$ then $\phi(x;\theta)_i = 0$.*

**(Symmetry):** *If $\forall \mathcal{S} \subset \mathcal{N} \backslash \{i,j\} \quad v_f(x_{\mathcal{S} \cup \{i\}}; \theta, \phi_0) = v_f(x_{\mathcal{S} \cup \{j\}}; \theta, \phi_0)$ then $\phi(x;\theta)_i = \phi(x;\theta)_j$.*

## 3 METHOD

The value function of feature subsets has inputs of variable sizes, i.e., $v_f : \bigcup_{k=1}^{N} \mathbb{R}^k \to \mathbb{R}$, whereas most existing models accept fix-size inputs, i.e., $f : \mathbb{R}^N \to \mathbb{R}$. Transforming from $f$ to $v_f$ demands approximating model output with missing inputs using handcrafted reference values (Lundberg & Lee (2017)) and a sampling procedure (Datta et al. (2016)). This is non-trivial given complex factors like feature dependency. SASANet addresses this by directly modeling $f : \bigcup_{k=1}^{N} \mathbb{R}^k \to \mathbb{R}$ using set-based modeling for inputs of any size.

**Definition 3.1 (SASANet value function)** *In SASANet, the value function $v_f(x_S; \theta, \phi_0)$ for any given feature subset $x_S \in \bigcup_{k=1}^{N} \mathbb{R}^k$ is the model output $f(x_S; \theta, \phi_0)$.*

With $v_f = f$, ensuring $f : \bigcup_{k=1}^{N} \mathbb{R}^k \to \mathbb{R}$ meets Definition 2.2 often involves adding a regularization term. This can cause conflicting optimization directions, hindering proper model convergence. To address this, we introduce an intermediate sequential modeling framework and an internal distillation strategy. This naturally achieves $f : \bigcup_{k=1}^{N} \mathbb{R}^k \to \mathbb{R}$ compliant with Definition 2.2. The process is depicted in Figure 1, with proofs and structural details in the Appendix.

### 3.1 MARGINAL CONTRIBUTION-BASED SEQUENTIAL MODELING

In SASANet, a marginal contribution-based sequential module generates a permutation-variant output for any feature set size, capturing the intermediate effects of each given feature. Specifically, for a given order $O$ where features in $\mathcal{N}$ are sequentially added for prediction, the marginal contribution of each feature is explicitly modeled as $\triangle(x_{O_i}, x_{O_{1:i-1}}; \theta_\triangle)$. For any feature subset $\mathcal{S} \subset \mathcal{N}$, accumulating the marginal contributions given an order $O_S \in \pi(\mathcal{S})$ yields the permutation-variant output $f_c(x_S, O_S; \theta_\triangle) = \sum_{i=1}^{|\mathcal{S}|} \triangle(x_{O_{S_i}}, x_{O_{S_{1:i-1}}}; \theta_\triangle) + \phi_0$. This module can be trained for various prediction tasks. For example, using $\sigma$ to represent the sigmoid function, we can formulate a binary classification loss for a sample $x$ as

$$L_m(x, y, O) = y \log(\sigma(f_c(x, O; \theta_\triangle))) + (1 - y) \log(1 - \sigma(f_c(x, O; \theta_\triangle))), \tag{3}$$

### 3.2 SHAPLEY VALUE DISTILLATION

Following Definition 2.1, we train an attribution network $\phi(\cdot; \theta_\phi) : \bigcup_{k=1}^{N} \mathbb{R}^k \to \bigcup_{k=1}^{N} \mathbb{R}^k$, producing a final permutation-invariant output $f(x_S; \theta_\phi) = \sum_{i \in \mathcal{S}} \phi(x_S; \theta_\phi)_i$ for any valid input $x_S \in \bigcup_{k=1}^{N} \mathbb{R}^k$. Specially, instead of directly supervising $f$ with data, we propose an internal distillation method based on the intermediate marginal contribution module $f_c$. Specifically, we construct a distillation loss for each feature $i \in \mathcal{S}$ in variable-size input $x_S \in \bigcup_{k=1}^{N} \mathbb{R}^k$ as

$$L_s^{(i)}(x_S) = \frac{1}{|\mathcal{D}|} \sum_{O \in \mathcal{D}} (\phi(x_S; \theta_\phi)_i - \sum_{k \in \mathcal{S}} \mathbb{I}\{O_k = i\} \triangle(x_i, x_{O_{1:k-1}}; \theta_\triangle))^2, \tag{4}$$

where $\mathcal{D} \subset \pi(\mathcal{S})$ denotes permutations drawn for training. Training with $L_s$, $\phi(x_S; \theta_\phi)_i$ amortizes the features' effect in $f_c$. We prove it naturally ensure Shapley value constraints without bringing conflict to optimization directions. For simplicity, as is typical, we assume no gradient vanishing and sufficient model expressiveness to reach optimal.

**Proposition 3.2** *By optimizing $L_s^{(i)}(x_S)$ with $\mathcal{D}$, SASANet converges to satisfy $\phi(x_S; \theta_\phi)_i \sim \mathcal{N}(\phi_i^*, \frac{\sigma_i^2}{M})$, where $M = |\mathcal{D}|$, $\phi_i^* = \frac{1}{|\mathcal{S}|!} \sum_{O \in \pi(\mathcal{S})} \sum_{k \in \mathcal{S}} \mathbb{I}\{O_k = i\} \triangle(x_i, x_{O_{1:k-1}}; \theta_\triangle)$, $\sigma_i^2 = \frac{1}{|\mathcal{S}|!} \sum_{O \in \pi(\mathcal{S})} \sum_{k \in \mathcal{S}} \mathbb{I}\{O_k = i\}(\triangle(x_i, x_{O_{1:k-1}}; \theta_\triangle) - \phi_i^*)^2$.*

This means $\phi$ is trained towards the averaged intermediate feature effects in $f_c$. Then, we can derive the relationship between the final output $f$ and the intermediate output $f_c$.

**Proposition 3.3** *By optimizing $L_s^{(i)}(x_S)$ with enough permutation sampling, there is small uncertainty that the model converges to satisfy $f(x_S; \theta_\phi) = \frac{1}{|\mathcal{S}|!} \sum_{O \in \pi(\mathcal{S})} f_c(x_S, O; \theta_\triangle)$.*

In this way, $f$ acts as an implicit bagging of the permutation-variant prediction of $f_c$, leading to valid permutation-invariant predictions that offer stability and higher accuracy, without requiring

direct supervision loss from training data. We will subsequently demonstrate how this distillation loss enables $\phi$ to model the Shapley value of $f$.

**Theorem 3.4** *If $\forall O_1, O_2 \in \pi(\mathcal{S})$ $f_c(x_\mathcal{S}, O_1) = f_c(x_\mathcal{S}, O_2)$, optimizing $L_s^{(i)}(x_\mathcal{S})$ for sample $x$'s subsets $x_\mathcal{S}$ with ample permutation samples ensures $\phi(x; \theta_\phi)$ converge to satisfy Definition 2.2's constraint, i.e., Shapley value in $f(x; \theta_\phi)$.*

While $f_c$ is permutation-variant, in the subsequent section, we'll introduce a method for $f_c$ to converge to label expectations of arbitrary feature subsets, thereby not only reflecting pertinent feature-label associations but also naturally inducing a permutation invariance condition.

## 3.3 FEATURE SUBSET LABEL EXPECTATION MODELING

Notably, training $f_c$ with Eq. 3 can lead to $f$ making good prediction for samples in the dataset, it does not guarantee meaningful outputs for feature subsets. For example, it might output 0 when any feature is missing, assigning the same Shapley value to all features as the last one takes all the credit. Although this attribution captures the model's logic, it fails to represent significant feature-label associations in the data. To address this, we target the output to reflect the label expectation for samples with a specific feature subset. Specifically, we define a loss for $x_\mathcal{S}$ using training set $D_{tr}$ as

$$L_v(x_\mathcal{S}, O_\mathcal{S}) = \frac{\sum_{(x', y') \in \mathcal{D}_{tr}} \mathbb{I}\{x'_\mathcal{S} = x_\mathcal{S}\} L_m(x_\mathcal{S}, y, O_\mathcal{S})}{\sum_{(x', y') \in \mathcal{D}_{tr}} \mathbb{I}\{x'_\mathcal{S} = x_\mathcal{S}\}}. \tag{5}$$

$L_v$ is designed for $f_c$ instead of the Shapley module, averting conflicts with the convergence direction of $\phi$ we have illustrated. Nonetheless, we show that this approach makes $\phi$ to be the Shapley value of $f$ by directing $f_c$'s output to satisfy the permutation-invariance stipulated in Theorem 3.4.

**Theorem 3.5** *Optimizing $L_s^{(i)}(x_\mathcal{S})$ and $L_v(x_\mathcal{S}, O_\mathcal{S})$ for sample $x$'s subsets $x_\mathcal{S}$ for all permutations $O_\mathcal{S} \in \pi(\mathcal{S})$ makes $\phi$ converge to Shapley value of $f$.*

In this way, we ensure the constraint in Definition 2.2 satisfied, while training $f$ to make a valid prediction with a unified distillation loss. Moreover, $L_v$ can make the attribution value seize real-world feature-label relevance, which we discuss as follows.

**Proposition 3.6** *Optimizing $L_s^{(i)}(x_\mathcal{S})$ and $L_v(x_\mathcal{S}, O_\mathcal{S})$ for all permutations $O_\mathcal{S} \in \pi(\mathcal{S})$ makes $\sigma(f(x_\mathcal{S}))$ converge to $\mathbb{E}_{\mathcal{D}_{tr}}[y|x_\mathcal{S}]$.*

While we cannot exhaust all the permutations in practice, by continuously sampling permutations during training, the network will learn to generalize by grasping permutation patterns.

**Remark 3.7** *$f$ estimates the expected label when certain features are observed, represented by $\int_{x'_{\bar{\mathcal{S}}}} p(x'_{\bar{\mathcal{S}}}|x_\mathcal{S}) h(x'_{\bar{\mathcal{S}}} \cup x_\mathcal{S}) dx'_{\bar{\mathcal{S}}}$, where $\bar{\mathcal{S}} = \mathcal{N} - \mathcal{S}$, $h(x)$ signifies a sample $x$'s real label, and $p(\cdot|x_\mathcal{S})$ is the conditional sample distribution given the feature set $x_\mathcal{S}$. Then, $\phi$ learns to estimate the actual feature-label Shapley value.*

Previously, such analyses can be conducted by training a model with fixed-size inputs and employing post-hoc methods. These methods crafted value functions to estimate expectations based on model outputs when randomly replacing missing features. As outlined in the literature (Štrumbelj & Kononenko (2014); Datta et al. (2016); Lundberg & Lee (2017)), this can be depicted as: $\int_{x'_{\bar{\mathcal{S}}}} p(x'_{\bar{\mathcal{S}}}) h'(x'_\mathcal{S} \cup x_{\bar{\mathcal{S}}}) dx'_{\bar{\mathcal{S}}}$, where $h'$ approximates the inaccessible $h$. The sample distribution for the entire dataset is given by $p_{gen}(x) = p(x_{\bar{\mathcal{S}}}) \cdot p(x_\mathcal{S})$, which only matches $p$ if $\forall x_{\bar{\mathcal{S}}}, x_\mathcal{S}$ $p(x_{\bar{\mathcal{S}}}) = p(x_{\bar{\mathcal{S}}}|x_\mathcal{S})$. Given that full feature independence is unlikely, generated samples may not align with real data, causing unreliable outputs. Despite progress in post-hoc studies (Laugel et al. (2019); Aas et al. (2021)) on feature dependence, complex tasks remain challenging. SASANet sidesteps this by learning an apt value function, better estimating feature-label relations via self-attribution.

## 3.4 POSITIONAL SHAPLEY VALUE DISTILLATION

The marginal contribution of a feature can fluctuate based on the prefix set size. For instance, with many observed features, a model might resist change in prediction from new ones. Such diversified

Table 1: Model performance - the best performance among interpretable models is in bold.

| Model | Income | | Higgs Boson | | Fraud | | Insurance | |
|---|---|---|---|---|---|---|---|---|
| | AP | AUC | AP | AUC | AP | AUC | RMSE | MAE |
| LightGBM | _0.6972_ | _0.9566_ | 0.8590 | 0.8459 | 0.7934 | 0.9737 | 0.2315 | _0.1052_ |
| MLP | 0.6616 | 0.9518 | _0.8877_ | _0.8771_ | _0.8167_ | _0.9621_ | _0.2310_ | 0.1076 |
| DT | 0.2514 | 0.7250 | 0.6408 | 0.6705 | 0.5639 | 0.8614 | 0.3332 | 0.1167 |
| LR | 0.3570 | 0.8717 | 0.6835 | 0.6846 | 0.7279 | 0.9620 | - | - |
| NAM | 0.6567 | 0.9506 | 0.7897 | 0.7751 | 0.7986 | 0.9590 | 0.2382 | 0.1182 |
| SITE | 0.6415 | 0.9472 | 0.8656 | 0.8597 | 0.7912 | 0.9556 | - | - |
| SENN | 0.6067 | 0.9416 | 0.7563 | 0.7556 | 0.7709 | 0.8916 | 0.2672 | 0.1313 |
| SASANet-p | 0.6708 | 0.9525 | 0.8775 | 0.8656 | 0.8090 | 0.9667 | **0.2368** | **0.0894** |
| SASANet-d | 0.6811 | 0.9527 | 0.8790 | 0.8675 | 0.8090 | 0.9665 | 0.2387 | 0.1037 |
| SASANet | **0.6864** | **0.9542** | **0.8836** | **0.8721** | **0.8124** | **0.9674** | 0.2375 | 0.0901 |

contribution distribution can hinder model convergence. Therefore, instead of directly training over-all attribution function with Eq. 4, we train a positional attribution function $\phi(x_{\mathcal{S}}; \theta_\phi)_{i,k}$ to measure feature $i$'s effects in $f_c$ in a certain position $k$, with an internal positional distillation loss:

$$L_s^{(i,k)}(x_{\mathcal{S}}) = \frac{1}{|\mathcal{D}|} \sum_{O \in \mathcal{D}} \mathbb{I}\{O_k = i\}(\phi(x_{\mathcal{S}}; \theta_\phi)_{i,k} - \triangle(x_i, x_{O_{1:k-1}}; \theta_\triangle))^2. \tag{6}$$

**Proposition 3.8** *By randomly drawing $m$ permutations where $x_i$ appears at a specific position $k$ and optimizing $L_s^{(i,k)}(x_{\mathcal{S}})$, we have $\phi(x_{\mathcal{S}}; \theta_\phi)_{i,k} \sim \mathcal{N}(\phi_{i,k}^*, \frac{\sigma_{i,k}^2}{m})$, where $\phi_{i,k}^* = \frac{1}{(|\mathcal{S}|-1)!} \sum_{O \in \pi(\mathcal{S})} \mathbb{I}\{O_k = i\}\triangle(x_i, x_{O_{1:k-1}}; \theta_\triangle)$, $\sigma_{i,k}^2 = \frac{1}{(|\mathcal{S}|-1)!} \sum_{O \in \pi(\mathcal{S})} \mathbb{I}\{O_k = i\}(\triangle(x_i, x_{O_{1:k-1}}; \theta_\triangle) - \phi_{i,k}^*)^2$.*

**Lemma 3.9** *When optimizing $L_s^{(i,k)}(x_{\mathcal{S}})$ with enough sampling and calculate $\phi(x_{\mathcal{S}}; \theta_\phi)_i = \frac{1}{|\mathcal{S}|} \sum_{i=1}^{|\mathcal{S}|} \phi(x_{\mathcal{S}}; \theta_\phi)_{i,k}$, there is small uncertainty that $\phi(x_{\mathcal{S}}; \theta_\phi)_i$ converges to $\frac{1}{|\mathcal{S}|!} \sum_{O \in \pi(\mathcal{S})} \sum_{k=1}^{|\mathcal{S}|} \mathbb{I}\{O_k = i\}\triangle(x_i, x_{O_{1:k-1}}; \theta_\triangle)$.*

The overall attribution value produced this way mirrors the original distillation, equating to the Shapley value in our analysis. Hence, we term it positional Shapley value distillation.

**Proposition 3.10** *Positional Shapley value distillation decreases SASANet's Shapley value estimation's variance.*

### 3.5 Network Implementation

To accommodate features of diverse meanings and distributions, we employ value embedding tables for categorical features and single-input MLPs for continuous ones, outperforming field embedding methods (Song et al. (2019); Guo et al. (2017)). The marginal contribution module applies multi-head attention for prefix representation of each feature, with position embeddings aiding in capturing sequential patterns and ensuring model convergence. Conversely, the Shapley value module uses multi-head attention for individualized sample representation, excluding position embeddings to maintain permutation-invariance. Feature embeddings are combined with prefix and sample representations for marginal contribution and positional Shapley value calculations using feed-forward networks. $\phi_0$ is precomputed, symbolizing the sample-specific label expectation. Refer to the Appendix for a model schematic. For computational efficiency, terms in $L_s$ and $L_v$ are separated based on sample and permutations, resulting in the loss formulation $L(x, y, O) = \sum_{k=1}^{N} (\lambda_s(\phi(x; \theta_\phi)O_k, k - \triangle(xO_k, x_{O_{1:k-1}}; \theta_\triangle))^2 + \lambda_v(\sum^k i = 1\triangle(xO_i, x_{O_{1:i-1}}; \theta_\triangle) + \phi_0 - y)^2)$. In this manner, for each sample in a batch, we randomly select a single permutation rather than sampling multiple times, promoting consistent training and enhancing efficiency.

## 4 Experiments

### 4.1 Experimental Setups

As depicted in Section 2.1, self-attributing models (including SASANet) can integrate pre-/post-transformations to adapt to various data types, e.g., integrate image feature extractors for image

concept-level attribution, then estimate pixel-level contributions with propagation methods like up-sampling. Yet, the efficacy and clarity of such attributions can be substantially compromised by the quality of the concept extraction module. To ensure clarity and sidestep potential ambiguities arising from muddled concepts or propagation methods, our evaluation uses tabular data, as it presents standardized and semantically coherent concepts. This approach foregrounds the direct impact of the core attribution layer. Specifically, we tested on three public classification tasks: Census Income Prediction (Kohavi (1996)), Higgs Boson Identification (Baldi et al. (2014)), and Credit Card Fraud Detection[2], and examined SASANet's regression on the Insurance Company Benchmark[3], all with pre-normalized input features. We benchmarked SASANet against prominent black-box models like LightGBM (Ke et al. (2017)) and MLP; traditional interpretable methods such as LR and DT; and other self-attributing networks like NAM, SITE, and SENN. Hyperparameters were finely tuned for each model on each dataset for a fair comparison, and models specific to classification weren't assessed on regression tasks. Configuration and dataset specifics are in Appendices H and I. For interpretation evaluation, we did not compare with other self-attributing models since their desired feature attribution value have different intuitions and physical meanings. Instead, we compared with post-hoc methods to demonstrate that SASANet accurately conveys its prediction rationale.

## 4.2 EFFECTIVENESS: PREDICTION PERFORMANCE EVALUATION

We used AUC and AP for imbalanced label classification, and RMSE and MAE for regression. Table 6 shows average scores from 10 tests; Appendix J lists standard deviations.

**Comparison to Baselines.** Black-box models outperform other compared models. Classic tree and linear methods, though interpretable, are limited by their simplicity. Similarly, current self-attributing networks often sacrifice performance for interpretability due to simplified structures or added regularizers. However, SASANet matches black-box model performance while remaining interpretable. We also show SASANet rivals separately trained MLP models in predicting with arbitrary number of missing features, indicating its Shapley values genuinely reflect feature-label relevance through modeling feature contribution to label expectation. See Appendix K for details.

**Ablation Study.** First, without the Shapley value module (i.e., "SASANet-d"), there was a large performance drop. This suggests that parameterizing Shapley value enhances accuracy. The permutation-variant predictions, despite theoretical convergence, can still be unstable in practice. The permutation-invariant Shapley value, akin to inherent bagging across permutations, offers better accuracy. Second, bypassing positional Shapley value and directly modeling overall Shapley value (i.e., "SASANet-p") resulted in performance reduction, particularly on Income and Higgs Boson datasets. This underscores the benefit of distinguishing marginal contributions at various positions.

## 4.3 FIDELITY: FEATURE-MASKING EXPERIMENT

We observed prediction performance after masking the top 1-5 features attributed by SASANet for each test sample and compared it to the outcome when using KernelSHAP, a popular post-hoc method, and FastSHAP, a recent parametric post-hoc method. The results are shown in Table 7. SASANet's feature masking leads to the most significant drop in performance, indicating its self-attribution is more faithful than post-hoc methods. This stems from self-attribution being rooted in real prediction logic. Moreover, SASANet's reliable predictions on varied feature subsets make it suitable for feature-masking experiments, sidestepping out-of-distribution noise from feature replacements. Notably, FastSHAP performs the worst in feature-masking experiments. While it can approximate the Shapley value in numerical regression, it appears to struggle in accurately identifying the most important features in a ranked manner. Additionally, we conducted experiment to add the top 1-5 features from scratch, showing SASANet's selection led to quicker performance improvement. See Appendix L for details.

## 4.4 EFFICIENCY: ATTRIBUTION TIME EVALUATION

We assessed attribution time for SASANet's self-attribution and two post-hoc interpreters, KernelSHAP and LIME, on 1,000 random samples. As per Table 3, SASANet was the quickest. LIME, reliant on drawing neighboring samples for local surrogates, was the slowest, especially with larger datasets. KernelSHAP, while faster than LIME due to its linear approximation and regression-based

---

[2]https://www.kaggle.com/datasets/mlg-ulb/creditcardfraud
[3]https://archive.ics.uci.edu/dataset/125/insurance+company+benchmark+coil+2000

Table 2: Results of feature masking experiments.

| Task | Method | Top 1 | | Top 2 | | Top 3 | | Top 4 | | Top 5 | |
|------|--------|-------|-----|-------|-----|-------|-----|-------|-----|-------|-----|
| | | AP | AUC | AP | AUC | AP | AUC | AP | AUC | AP | AUC |
| Income | SASA. | **0.560** | **0.929** | **0.497** | **0.917** | **0.441** | **0.904** | **0.398** | **0.892** | **0.361** | **0.880** |
| | KerSH. | 0.606 | 0.939 | 0.566 | 0.932 | 0.547 | 0.928 | 0.532 | 0.926 | 0.518 | 0.923 |
| | FastSH. | 0.683 | 0.954 | 0.617 | 0.944 | 0.617 | 0.944 | 0.618 | 0.944 | 0.613 | 0.943 |
| Higgs | SASA. | **0.825** | **0.813** | **0.777** | **0.764** | **0.730** | **0.715** | **0.681** | **0.665** | **0.632** | **0.616** |
| | KerSH. | 0.855 | 0.843 | 0.833 | 0.821 | 0.808 | 0.795 | 0.783 | 0.770 | 0.760 | 0.746 |
| | FastSH. | - | - | - | - | - | - | - | - | - | - |
| Fraud | SASA. | **0.789** | **0.962** | **0.758** | **0.957** | **0.681** | **0.952** | **0.625** | **0.938** | **0.526** | **0.904** |
| | KerSH. | 0.806 | 0.963 | 0.782 | 0.961 | 0.732 | 0.959 | 0.693 | 0.949 | 0.627 | 0.936 |
| | FastSH. | 0.813 | 0.967 | 0.813 | 0.967 | 0.815 | 0.965 | 0.811 | 0.965 | 0.809 | 0.966 |

[*] FastSHAP results for the Higgs dataset are missing due to prolonged training times.

estimation, was constrained by its sampling, making it over 200 times slower than SASANet. Notably, while parametric post-hoc methods like FastSHAP (Jethani et al. (2021)) offer swift 1-pass attribution, they demand an extensive post-hoc training involving many forward propagations of the interpreted model. This was problematic for large datasets and models, such as the Higgs Boson. Conversely, self-attribution itself is the core component of model prediction and inherently acquired during model training, eliminating post-hoc sampling or regression procedure.

## 4.5 ACCURACY: COMPARISON WITH GROUND-TRUTH SHAPLEY VALUE

SASANet can predict for any-sized input, allowing us to directly ascertain accurate Shapley values through sufficient permutations. While getting ground truth Shapley value is time-intensive, the transformer structure in SASANet alleviates the problem since it can simultaneously produce multiple feature subsets' value by incrementally considering the attention relevant to new features. For each dataset, we sampled 1,000 test samples and estimated their real Shapley values in SASANet with 10,000 permutations. Furthermore, to test the generalization of the attribution module, we created a distribution shift dataset by adding a randomly sampled large bias to each dimension, which has the same scale as the normalized input. The RMSE of KernelSHAP, FastSHAP, and SASANet's attribution value to the real Shapley value are shown in Table 3. We have observed that SASANet's self-attribution models provide accurate estimates of the Shapley value for its own output. The performance of the attribution module may decrease when there is a distribution shift, as it is trained on the distribution of the training data in our experiment. Nevertheless, it still largely outperforms KernelSHAP and FastSHAP. Indeed, our analysis in Section 3.2 indicates that convergence to the Shapley value is ensured on the distilled sample distribution, irrespective of input distribution. Therefore, simple tricks like introducing noise during internal distillation can alleviate this issue without harming the effectiveness of the model.

Table 3: Time cost and accuracy for feature attribution.

| Method | Time | | | RMSE | | | RMSE (Distribution Shift) | | |
|--------|--------|-------|-------|--------|-------|-------|--------|-------|-------|
| | Income | Higgs | Fraud | Income | Higgs | Fraud | Income | Higgs | Fraud |
| LIME | 1604.9s | 11792.7s | 9851.2s | - | - | - | - | - | - |
| KernelSHAP | 91.2s | 65.7s | 38.4s | 0.348 | 0.402 | 0.514 | 0.348 | 0.405 | 0.494 |
| FastSHAP[*] | **0.3s**[*] | -[*] | **0.2s**[*] | 0.332 | - | 0.473 | 0.522 | - | 0.550 |
| SASANet | **0.3s** | **0.3s** | **0.2s** | **0.001** | **0.005** | **0.033** | **0.001** | **0.113** | **0.082** |

[*] FastSHAP's has time-consuming post-hoc training procedure, whose time isn't reported to prevent confusion. FastSHAP results for the Higgs dataset are missing due to prolonged training times.

## 4.6 QUALITATIVE EVALUATION

**Overall Interpretation.** We compared feature attributions from SASANet and KernelSHAP for the entire test set. In Figure 2 (a), SASANet reveals clear attribution patterns. Specifically, for the Income dataset, there's evident clustering in attribution values tied to specific feature values. Such sparsity of attribution value aligns with the feature value, which are mostly categorical. The Higgs Boson dataset shows SASANet's attributions tend to align with feature value changes, confirming its ability to identify pertinent feature-label relationships. For Higgs Boson dataset, SASANet's attribution value shows rough monotonic associations with feature values. For example, a larger feature value tends to bring positive attribution for "m_jlv" while a negative attribution value for "m_bb". This suggests SASANet identifies feature-label relevance consistent with common sense. That is, despite complex quantitative relationship between features and outcomes, qualitative patterns are

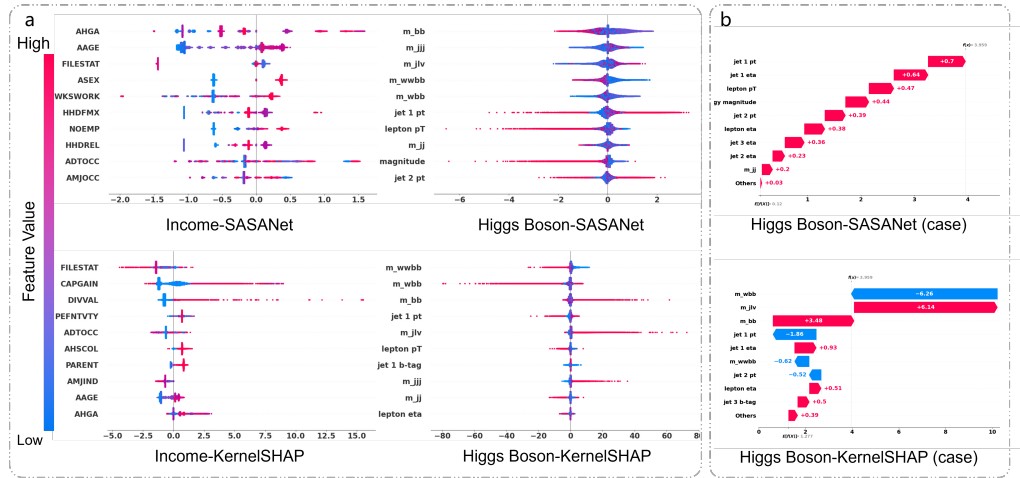

Figure 2: Feature attribution visualizations: x-axis shows attribution value; y-axis lists top features by decreasing average absolute attribution. (a) Overall attribution. (b) Single-instance attribution.

usually clear. Meanwhile, clear differences exist between real self-attribution and approximated post-hoc attribution, highlighting potential inaccuracies in post-hoc methods. For example, in both datasets, many features have abnormally high attribution values in KernelSHAP. This may stem from neglecting feature interdependencies, evident upon sample interpretation in the subsequent part.

**Sample Interpretation.** We randomly draw a positive sample in Higgs Boson dataset predicted correctly by SASANet and compared its self-attribution with KernelSHAP's attribution in Figure 2 (b). Distinct differences are evident. Notably, SASANet indicates all top-9 features contribute positively, highlighting a simple, intuitive logic. Indeed, the objective of Higgs Boson prediction is to differentiate processes that produce Higgs bosons from those that do not. As a result, the model distinguishes the distinctive traits of Higgs bosons from the ordinary, resulting in a consistent positive influence from key features representing these unique attributes. KernelSHAP sometimes assigns disproportionately large negative values to features. For instance, the attribution for "m_wbb" ($-6.26$) significantly surpasses SASANet's peak attribution of $0.7$. Yet, they aggregate to a final output of $f(x) = 3.959$ as opposing attributions from dependent features, like the positive contribution of "m_jlv" ($6.14$), neutralize each other. Notably, SASANet's self-attribution doesn't deem either m_wbb" or "m_jlv" as so crucial. Similar phenomenon can be observed from other datasets and samples, which are visualized in Appendix M. For example, in Income prediction task, indicators exist for both rich and poor people. Accordingly, SASANet identifies input features indicating both outcomes. However, it is still obvious that KernelSHAP attributes exaggerated values to features, which then offset one another. This underscores how KernelSHAP can be deceptively complex by neglecting feature interdependencies, an issue that has been a pitfall in numerous post-hoc studies (Laugel et al. (2019); Frye et al. (2020); Aas et al. (2021)). SASANet naturally handles feature dependency well since the value functions has explicit physical meaning as estimated label expectation and are directly trained under the real data distribution.

## 5 CONCLUDING REMARKS

We proposed Shapley Additive Self-Attributing Neural Network (SASANet) and proved its self-attribution aligns with the Shapley values of its attribution-generated output. Through extensive experiments on real-world datasets, we demonstrated that SASANet not only surpasses current self-interpretable models in performance but also rivals the precision of black-box models while maintaining faithful Shapley attribution, bridging the gap between expressiveness and interpretability. Furthermore, we showed that SASANet excels in interpreting itself compared to using post-hoc methods. Moreover, our theoretical analysis and experiments suggest broader applications for SASANet. Firstly, its self-attribution offers a foundation for critiquing post-hoc interpreters, pinpointing areas of potential misinterpretation. Furthermore, its focus on label expectation uncovers intricate non-linear relationships between real-world features and outcomes. In the future, we will use SASANet to gain further insights into model interpretation and knowledge discovery studies.

## 6 ETHICS STATEMENT

This is a fundamental research on a neural network model that is not tied to a particular application. All the data used in this study are popular public datasets. The authors assert that there are no potential ethical concerns associated with this paper.

## 7 REPRODUCIBILITY STATEMENT

The code for this paper has been uploaded to https://anonymous.4open.science/r/SASANet-B343 for peer review and will be publicly available after publication. The data used in this paper are popular public datasets, and we have provided links or citations to their sources. All the proofs of theoretical results have been appended in the appendix.

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

# A    PROOF OF PROPOSITION 3.2

Given that $L_s^{(i)}(x_{\mathcal{S}}) = \frac{1}{|\mathcal{D}|}\sum_{O \in \mathcal{D}}(\phi(x_{\mathcal{S}};\theta_\phi)_i - \sum_{k \in \mathcal{S}}\mathbb{I}\{O_k = i\}\triangle(x_i, x_{O_{1:k-1}};\theta_\triangle))^2$, the partial derivative of $L_s^{(i)}(x, y)$ with respective to $\theta_\phi$ can be calculated as

$$\frac{\partial L_s^{(i)}(x_{\mathcal{S}})}{\partial \theta_\phi} = \frac{2}{|\mathcal{D}|}\sum_{O \in \mathcal{D}}(\phi(x_{\mathcal{S}};\theta_\phi)_i - \sum_{k=1}^{N}\mathbb{I}\{O_k = i\}\triangle(x_i, x_{O_{1:k-1}};\theta_\triangle))\frac{\partial \phi(x_{\mathcal{S}};\theta_\phi)_i}{\partial \theta_\phi}. \quad (7)$$

The loss function converges when $\frac{L_s^{(i)}(x_{\mathcal{S}})}{\partial \theta_\phi} = 0$. When $\frac{\partial \phi(x_{\mathcal{S}};\theta_\phi)_i}{\partial \theta_\phi} \neq 0$, we have

$$\sum_{O \in \mathcal{D}}\phi(x_{\mathcal{S}};\theta_\phi)_i = \sum_{O \in \mathcal{D}}\sum_{k=1}^{N}\mathbb{I}\{O_k = i\}\triangle(x_i, x_{O_{1:k-1}};\theta_\triangle). \quad (8)$$

Then, we can derive

$$\phi(x_{\mathcal{S}};\theta_\phi)_i = \frac{1}{|\mathcal{D}|}\sum_{O \in \mathcal{D}}\sum_{k=1}^{N}\mathbb{I}\{O_k = i\}\triangle(x_i, x_{O_{1:k-1}};\theta_\triangle). \quad (9)$$

Therefore, we have

**Proposition A.1** *By optimizing $L_s^{(i)}(x_{\mathcal{S}})$, unless gradient vanishing (i.e., $\frac{\partial \phi(x_{\mathcal{S}};\theta_\phi)_i}{\partial \theta_\phi} \neq 0$), an expressive enough model converges to $\phi(x_{\mathcal{S}};\theta_\phi)_i = \frac{1}{|\mathcal{D}|}\sum_{O \in \mathcal{D}}\sum_{k \in \mathcal{S}}\mathbb{I}\{O_k = i\}\triangle(x_i, x_{O_{1:k-1}};\theta_\triangle)$.*

According to Proposition A.1, optimizing $L_s^{(i)}(x_{\mathcal{S}})$ when $|\mathcal{D}| = M$ will make $\phi(x_{\mathcal{S}};\theta_\phi)_i$ converge to

$$\phi(x_{\mathcal{S}};\theta_\phi)_i = \frac{1}{M}\sum_{O \in \mathcal{D}}\sum_{k=1}^{|\mathcal{S}|}\mathbb{I}\{O_k = i\}\triangle(x_i, x_{O_{1:k-1}};\theta_\triangle), \quad (10)$$

i.e., the averaged marginal contribution in the drawn permutations.

Since the permutations are sampled randomly and independently, their corresponding marginal contribution are i.i.d. We can regard them as randomly sampled from a marginal contribution set

$$\mathcal{D}_\triangle^i(x_{\mathcal{S}}) := \{\sum_{k=1}^{|\mathcal{S}|}\mathbb{I}\{O_k = i\}\triangle(x_i, x_{O_{1:k-1}};\theta_\triangle)|O \in \pi(\mathcal{S})\}. \quad (11)$$

According to Central Limit Theorem, their mean follows a Gaussian distribution

$$\mathcal{N}(\phi_i^*, \frac{\sigma_i^2}{M}), \quad (12)$$

where $\phi_i^* = \frac{1}{|\mathcal{S}|!}\sum_{\triangle \in \mathcal{D}_\triangle^i(x_{\mathcal{S}})}\triangle$ is the real Shapley value, $\sigma_i^2 = \frac{1}{|\mathcal{S}|!}\sum_{\triangle \in \mathcal{D}_\triangle^i(x_{\mathcal{S}})}(\triangle - \phi_i^*)^2$ is the variance of $\mathcal{D}_\triangle^i(x_{\mathcal{S}})$. Obviously, we can formulate $\phi_i^*$ and $\sigma_i^2$ as

$$\phi_i^* = \frac{1}{|\mathcal{S}|!}\sum_{O \in \pi(\mathcal{S})}\sum_{k=1}^{|\mathcal{S}|}\mathbb{I}\{O_k = i\}\triangle(x_i, x_{O_{1:k-1}};\theta_\triangle),$$

$$\sigma_i^2 = \frac{1}{|\mathcal{S}|!}\sum_{O \in \pi(\mathcal{S})}(\sum_{k=1}^{|\mathcal{S}|}\mathbb{I}\{O_k = i\}\triangle(x_i, x_{O_{1:k-1}};\theta_\triangle) - \phi_i^*)^2. \quad (13)$$

Therefore, $\phi(x_{\mathcal{S}};\theta_\phi)_i \sim \mathcal{N}(\phi_i^*, \frac{\sigma_i^2}{M})$, where $\phi_i^* = \frac{1}{|\mathcal{S}|!}\sum_{O \in \pi(\mathcal{S})}\sum_{k=1}^{|\mathcal{S}|}\mathbb{I}\{O_k = i\}\triangle(x_i, x_{O_{1:k-1}};\theta_\triangle)$, $\sigma_i^2 = \frac{1}{|\mathcal{S}|!}\sum_{O \in \pi(\mathcal{S})}\sum_{k=1}^{|\mathcal{S}|}\mathbb{I}\{O_k = i\}(\triangle(x_i, x_{O_{1:k-1}};\theta_\triangle) - \phi_i^*)^2$.

## B  PROOF OF PROPOSITION 3.3

We can derive from Proposition 3.2 that

**Lemma B.1** *When optimizing $L_s^{(i)}(x_\mathcal{S})$ with enough permutation sampling, there is small uncertainty that $\phi(x_\mathcal{S}; \theta_\phi)_i$ converges to $\frac{1}{|\mathcal{S}|!} \sum_{O \in \pi(\mathcal{S})} \sum_{k \in \mathcal{S}} \mathbb{I}\{O_k = i\} \triangle(x_i, x_{O_{1:k-1}}; \theta_\triangle)$.*

According to Lemma B.1, for any $\mathcal{S} \in \mathcal{N}$, it holds that $\phi_0 + \sum_{i \in \mathcal{S}} \phi_i(x_\mathcal{S}; \theta_\phi) = \frac{1}{|\mathcal{S}|!} \sum_{O \in \pi(\mathcal{S})} \sum_{k=1}^{|\mathcal{S}|} \triangle(x_k, x_{O_{1:k-1}}; \theta_\triangle) = \frac{1}{|\mathcal{S}|!} \sum_{O_\mathcal{S} \in \pi(\mathcal{S})} f_c(x_\mathcal{S}, O_\mathcal{S})$. Since $f(x_\mathcal{S}) = \phi_0 + \sum_{i \in \mathcal{S}} \phi_i(x_\mathcal{S}; \theta_\phi)$, we have $f(x_\mathcal{S}) = \frac{1}{|\mathcal{S}|!} \sum_{O \in \pi(\mathcal{S})} f_c(x_\mathcal{S}, O)$.

## C  PROOF OF THEOREM 3.4

According to Proposition 3.3, by optimizing $L_s^{(i)}(x_\mathcal{S})$, we have $f(x_\mathcal{S}) = \frac{1}{|\mathcal{S}|!} \sum_{O \in \pi(\mathcal{S})} f_c(x_\mathcal{S}, O)$. When $f_c(x_\mathcal{S}, O_1) = f_c(x_\mathcal{S}, O_2)$ $\forall O_1, O_2 \in \pi(\mathcal{S})$, it is obvious that $f(x_\mathcal{S}) = f_c(x_\mathcal{S}, O)$ $\forall O \in \pi(\mathcal{S})$. According to Lemma B.1,

$$\phi_i(x; \theta_\phi) = \frac{1}{N!} \sum_{O \in \pi(\mathcal{N})} \sum_{k=1}^{N} \mathbb{I}\{O_k = i\} \triangle(x_i, x_{O_{1:k-1}}; \theta_\triangle)$$

$$= \frac{1}{N!} \sum_{O \in \pi(\mathcal{N})} \sum_{k=1}^{N} \mathbb{I}\{O_k = i\}(f_c(x_{O_{1:k}}, O_{1:k}) - f_c(x_{O_{1:k-1}}, O_{1:k-1})) \quad (14)$$

$$= \frac{1}{N!} \sum_{O \in \pi(\mathcal{N})} \sum_{k=1}^{N} \mathbb{I}\{O_k = i\}(f(x_{O_{1:k}}) - f(x_{O_{1:k-1}})).$$

In this case, according to Shapley et al. (1953), $\phi$ holds the four axioms and is the Shapley value of $f$.

## D  PROOF OF THEOREM 3.5

First, we induce the value of $f_c$ when it converges.

**Proposition D.1** *Optimizing $L_v(x_\mathcal{S}, O_\mathcal{S})$ makes $f_c(x_\mathcal{S}, O_\mathcal{S})$ converge to $\sigma^{-1}(\mathbb{E}_{\mathcal{D}_{tr}}[y|x_\mathcal{S}])$.*

We can prove it as follows. Given

$$L_v(x_\mathcal{S}, O_\mathcal{S}) = \frac{\sum_{(x',y') \in \mathcal{D}_{tr}} \mathbb{I}\{x'_\mathcal{S} = x_\mathcal{S}\} L_m(x_\mathcal{S}, y, O_\mathcal{S})}{\sum_{(x',y') \in \mathcal{D}_{tr}} \mathbb{I}\{x'_\mathcal{S} = x_\mathcal{S}\}}, \quad (15)$$

the partial derivative of $L_v(x_\mathcal{S}, y, O_\mathcal{S})$ with respective to $\theta_\phi$ can be derived as

$$\frac{\partial L_v(x_\mathcal{S}, y, O_\mathcal{S})}{\partial \theta_\phi} = \frac{1}{\sum_{(x',y') \in \mathcal{D}_{tr}} \mathbb{I}\{x'_\mathcal{S} = x_\mathcal{S}\}} \sum_{(x',y') \in \mathcal{D}_{tr}} \mathbb{I}\{x'_\mathcal{S} = x_\mathcal{S}\} \frac{\partial L_m(x_\mathcal{S}, y, O_\mathcal{S})}{\partial \theta_\phi}. \quad (16)$$

The partial derivative of $L_m(x_\mathcal{S}, y, O_\mathcal{S}) = y \log(\sigma(f_c(x_\mathcal{S}, O_\mathcal{S}))) + (1-y) \log(1 - \sigma(f_c(x_\mathcal{S}, O_\mathcal{S})))$ with respective to $\hat{y} = \sigma(f_c(x_\mathcal{S}, O_\mathcal{S}))$ can be derived as

$$\frac{\partial L_m(x_\mathcal{S}, y, O_\mathcal{S})}{\partial \hat{y}} = \frac{\hat{y} - y}{\hat{y}(1 - \hat{y})}.$$

The partial derivative of $\hat{y} = \sigma(f_c(x_\mathcal{S}, O_\mathcal{S}))$ with respective to $f_c(x_\mathcal{S}, O_\mathcal{S})$ can be derived as

$$\frac{\partial \hat{y}}{\partial f_c(x_\mathcal{S}, O_\mathcal{S})} = \hat{y}(1 - \hat{y}).$$

Therefore, the partial derivative of $L_m(x_{\mathcal{S}}, y, O_{\mathcal{S}})$ with respective to $\theta_\phi$ can be derived as

$$\frac{\partial L_m(x_{\mathcal{S}}, y, O_{\mathcal{S}})}{\partial \theta_\phi} = (\hat{y} - y)\frac{\partial f_c(x_{\mathcal{S}}, O_{\mathcal{S}})}{\partial \theta_\phi}.$$

Therefore, we have

$$\frac{\partial L_v(x_{\mathcal{S}}, y, O_{\mathcal{S}})}{\partial \theta_\phi} = \frac{1}{\sum_{(x', y') \in \mathcal{D}_{tr}} \mathbb{I}\{x'_{\mathcal{S}} = x_{\mathcal{S}}\}}\frac{\partial f_c(x_{\mathcal{S}}, O_{\mathcal{S}})}{\partial \theta_\phi} \sum_{(x', y') \in \mathcal{D}_{tr}} \mathbb{I}\{x'_{\mathcal{S}} = x_{\mathcal{S}}\}(\hat{y} - y'). \tag{17}$$

The loss function converges when $\frac{\partial L_v(x_{\mathcal{S}}, y, O_{\mathcal{S}})}{\partial \theta_\phi} = 0$. When $\frac{\partial f_c(x_{\mathcal{S}}, O_{\mathcal{S}})}{\partial \theta_\phi} \neq 0$, i.e., gradient descent not occurring for the prediction of the marginal contribution module, we have

$$\sum_{(x', y') \in \mathcal{D}_{tr}} \mathbb{I}\{x'_{\mathcal{S}} = x_{\mathcal{S}}\}\hat{y} = \sum_{(x', y') \in \mathcal{D}_{tr}} \mathbb{I}\{x'_{\mathcal{S}} = x_{\mathcal{S}}\}y' \tag{18}$$

Then, we can derive

$$\sigma(f_c(x_{\mathcal{S}}, O_{\mathcal{S}})) = \frac{\sum_{(x', y') \in \mathcal{D}_{tr}} \mathbb{I}\{x'_{\mathcal{S}} = x_{\mathcal{S}}\}y'}{\sum_{(x', y') \in \mathcal{D}_{tr}} \mathbb{I}\{x'_{\mathcal{S}} = x_{\mathcal{S}}\}} = \mathbb{E}_{\mathcal{D}_{tr}}[y|x_{\mathcal{S}}]. \tag{19}$$

Therefore, $f_c(x_{\mathcal{S}}, O_{\mathcal{S}}) = \sigma^{-1}(\mathbb{E}_{\mathcal{D}_{tr}}[y|x_{\mathcal{S}}])$.

According to Proposition D.1, for all $O \in \pi(\mathcal{S})$, there is $\sigma(f_c(x_{\mathcal{S}}, O_{\mathcal{S}})) = \mathbb{E}_{\mathcal{D}_{tr}}[y|x_{\mathcal{S}}]$. According to Theorem 3.6, this leads to $\phi$ converging to Shapley value of $f$.

## E    PROOF OF PROPOSITION 3.6

According to Proposition 3.3, $f(x_{\mathcal{S}}) = \frac{1}{|\mathcal{S}|!}\sum_{O \in \pi(\mathcal{S})} f_c(x_{\mathcal{S}}, O; \theta_\delta)$. With Proposition D.1, we have $f(x_{\mathcal{S}}) = \frac{1}{|\mathcal{S}|!}\sum_{O \in \pi(\mathcal{S})} \sigma^{-1}(\mathbb{E}_{\mathcal{D}_{tr}}[y|x_{\mathcal{S}}]) = \sigma^{-1}(\mathbb{E}_{\mathcal{D}_{tr}}[y|x_{\mathcal{S}}])$. Therefore, $\sigma(f(x_{\mathcal{S}})) = \mathbb{E}_{\mathcal{D}_{tr}}[y|x_{\mathcal{S}}]$.

## F    PROOF OF PROPOSITION 3.8

We first prove the following proposition:

**Proposition F.1** *Optimizing* $L_s^{(i,k)}(x_{\mathcal{S}})$ *makes* $\phi(x_{\mathcal{S}}; \theta_\phi)_{i,k}$ *converge* *to* $\frac{\sum_{O \in \mathcal{D}} \mathbb{I}\{O_k = i\}\triangle(x_i, x_{O_{1:k-1}}; \theta_\triangle)}{\sum_{O \in \mathcal{D}} \mathbb{I}\{O_k = i\}}$.

Given

$$L_s^{(i,k)}(x_{\mathcal{S}}) = \frac{1}{|\mathcal{D}|}\sum_{O \in \mathcal{D}} \mathbb{I}\{O_k = i\}(\phi(x_{\mathcal{S}}; \theta_\phi)_{i,k} - \triangle(x_i, x_{O_{1:k-1}}; \theta_\triangle))^2, \tag{20}$$

the partial derivative of $L_s^{(i,k)}(x, y)$ with respective to $\theta_\phi$ can be calculated as

$$\frac{\partial L_s^{(i,k)}(x_{\mathcal{S}})}{\partial \theta_\phi} = \frac{2}{|\mathcal{D}|}\sum_{O \in \mathcal{D}} \mathbb{I}\{O_k = i\}(\phi(x_{\mathcal{S}}; \theta_\phi)_{i,k} - \triangle(x_i, x_{O_{1:k-1}}; \theta_\triangle))\frac{\partial \phi(x_{\mathcal{S}}; \theta_\phi)_{i,k}}{\partial \theta_\phi}. \tag{21}$$

The loss function converges when $\frac{\partial L_s^{(i,k)}(x_{\mathcal{S}})}{\partial \theta_\phi} = 0$. When $\frac{\partial \phi(x_{\mathcal{S}}; \theta_\phi)_{i,k}}{\partial \theta_\phi} \neq 0$, we have

$$\sum_{O \in \mathcal{D}} \mathbb{I}\{O_k = i\}(\phi(x_{\mathcal{S}}; \theta_\phi)_{i,k} - \triangle(x_i, x_{O_{1:k-1}}; \theta_\triangle)) = 0. \tag{22}$$

From Eq. 22, we can derive

$$\phi(x_{\mathcal{S}}; \theta_\phi)_{i,k} = \frac{\sum_{O \in \mathcal{D}} \mathbb{I}\{O_k = i\}\triangle(x_i, x_{O_{1:k-1}}; \theta_\triangle)}{\sum_{O \in \mathcal{D}} \mathbb{I}\{O_k = i\}}. \tag{23}$$

According to Proposition A.1, optimizing $L_s^{(i,k)}(x_\mathcal{S})$ will make $\phi(x_\mathcal{S};\theta_\phi)_{i,k}$ converge to $\frac{\sum_{O\in\mathcal{D}}\mathbb{I}\{O_k=i\}\triangle(x_i,x_{O_{1:k-1}};\theta_\triangle)}{\sum_{O\in\mathcal{D}}\mathbb{I}\{O_k=i\}}$. i.e., the averaged marginal contribution in permutations that $x_i$ appears at the $k$-$th$ position. Supposing $m = \sum_{O\in\mathcal{D}}\mathbb{I}\{O_k=i\}$, since the permutations are drawn independently from each other, their generated marginal contribution are i.i.d. We can regard them as randomly sampled from the marginal contribution set

$$\mathcal{D}_\triangle^{i,k}(x_\mathcal{S}) = \{\triangle(x_i,x_{O_{1:k-1}};\theta_\triangle)|O\in\pi(\mathcal{S}),O_k=i\}. \tag{24}$$

According to Central Limit Theorem, their mean value follows a Gaussian distribution:

$$\mathcal{N}(\phi_{i,k}^*, \frac{\sigma_{i,k}^2}{m}), \tag{25}$$

where $\phi_{i,k}^* = \frac{1}{(|\mathcal{S}|-1)!}\sum_{\triangle\in\mathcal{D}_\triangle^{i,k}(x_i)}\triangle$ is the real positional Shapley value, $\sigma_{i,k}^2 = \frac{1}{(|\mathcal{S}|-1)!}\sum_{\triangle\in\mathcal{D}_\triangle^{i,k}(x_i)}(\triangle-\phi_{i,k}^*)^2$ is the variance of $\mathcal{D}_\triangle$. Obviously, we can formulate $\phi_{i,k}^*$ and $\sigma_{i,k}^2$ as

$$\phi_{i,k}^* = \frac{1}{(|\mathcal{S}|-1)!}\sum_{O\in\pi(\mathcal{S})}\mathbb{I}\{O_k=i\}\triangle(x_i,x_{O_{1:k-1}};\theta_\triangle)$$

$$\sigma_{i,k}^2 = \frac{1}{(|\mathcal{S}|-1)!}\sum_{O\in\pi(\mathcal{S})}\mathbb{I}\{O_k=i\}(\triangle(x_i,x_{O_{1:k-1}};\theta_\triangle)-\phi_{i,k}^*)^2. \tag{26}$$

Therefore, $\phi_{i,k}\sim\mathcal{S}(\phi_{i,k}^*,\frac{\sigma_{i,k}^2}{m})$, where $\phi_{i,k}^* = \frac{1}{(|\mathcal{S}|-1)!}\sum_{O\in\pi(\mathcal{S})}\mathbb{I}\{O_k=i\}\triangle(x_i,x_{O_{1:k-1}};\theta_\triangle)$, $\sigma_{i,k}^2 = \frac{1}{(|\mathcal{S}|-1)!}\sum_{O\in\pi(\mathcal{S})}\mathbb{I}\{O_k=i\}(\triangle(x_i,x_{O_{1:k-1}};\theta_\triangle)-\phi_{i,k}^*)^2$.

## G    PROOF OF PROPOSITION 3.10

Since the position of feature $x_i$ in the sampled permutation follows a uniform distribution, when sampling large number of permutations, the number of samples appearing at different positions are approximately the same. We denote the number as $m = M/|\mathcal{S}|$. Since

$$\phi(x_\mathcal{S};\theta_\phi)_{i,k}\sim\mathcal{N}(\phi_{i,k}^*,\frac{\sigma_{i,k}^2}{m}), \tag{27}$$

When using positional Shapley value to estimate Shapley value, we have

$$\phi(x_\mathcal{S};\theta_\phi)_i = \frac{1}{|\mathcal{S}|}\sum_{k=1}^{|\mathcal{S}|}\phi(x_\mathcal{S};\theta_\phi)_{i,k}$$

$$\sim\mathcal{N}(\frac{\sum_{k=1}^{|\mathcal{S}|}\phi_{i,k}^*}{|\mathcal{S}|},\frac{1}{|\mathcal{S}|^2}\sum_{k=1}^{|\mathcal{S}|}\frac{\sigma_k^2}{m}) \tag{28}$$

$$= \mathcal{N}(\phi_i^*,\frac{1}{|\mathcal{S}|^2}\sum_{k=1}^{N}\frac{\sigma_k^2}{m}).$$

According to Proposition 3.2, the distribution of directly Shapley value estimation is $\mathcal{N}(\phi_i^*,\frac{\sigma_i^2}{M})$. Compare the variances of these two distributions, we have

$$\frac{\sigma_i^2}{M}-\frac{1}{|\mathcal{S}|^2}\sum_{k=1}^{|\mathcal{S}|}\frac{\sigma_{i,k}^2}{m}\approx\frac{\sigma_i^2}{M}-\frac{\sum_{k=1}^{|\mathcal{S}|}\sigma_{i,k}^2}{|\mathcal{S}|M}$$

$$= \frac{1}{|\mathcal{S}|M}(|\mathcal{S}|\sigma_i^2-\sum_{k=1}^{|\mathcal{S}|}\sigma_{i,k}^2) \tag{29}$$

$$= \frac{1}{|\mathcal{S}|M}(\sum_{k=1}^{|\mathcal{S}|}(\sigma_{i,k}')^2-\sum_{k=1}^{|\mathcal{S}|}\sigma_{i,k}^2),$$

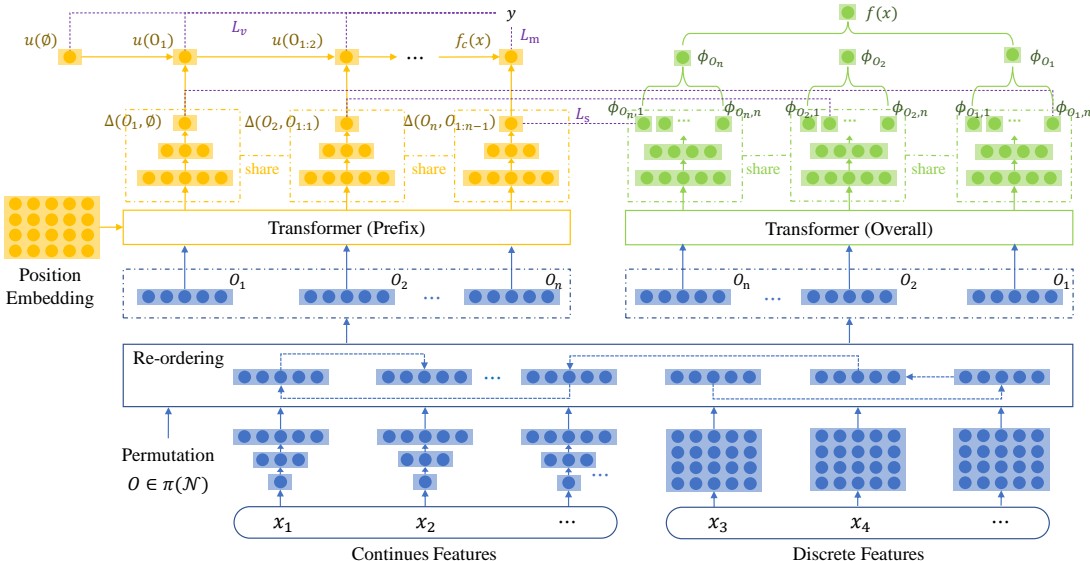

Figure 3: **The structure overview of SASANet.** Different modules are represented by different colors, including Feature Embedding module (blue), Marginal Contribution module (yellow), and Shapley Value module (green). The purple dash lines indicate loss terms.

where

$$(\sigma'_{i,k})^2 = \frac{\sum_{O \in \pi(\mathcal{S})} \mathbb{I}\{O_k = i\}(\triangle(x_i, x_{O_{1:k}}; \theta_\triangle) - \phi_i^*)^2}{(|\mathcal{S}| - 1)!}. \tag{30}$$

Since

$$\phi_{i,k}^* = \arg\min_\mu \sum_{O \in \pi(\mathcal{S})} \mathbb{I}\{O_k = i\}(\triangle(x_i, x_{O_{1:k}}; \theta_\triangle) - \mu)^2, \tag{31}$$

we have

$$\begin{aligned}
\sigma_{i,k}^2 &= \frac{\sum_{O \in \pi(\mathcal{S})} \mathbb{I}\{O_k = i\}(\triangle(x_i, x_{O_{1:k-1}}; \theta_\triangle) - \phi_{i,k})^2}{(|\mathcal{S}| - 1)!} \\
&= \frac{\min_\mu \sum_{O \in \pi(\mathcal{S})} \mathbb{I}\{O_k = i\}(\triangle(x_i, x_{O_{1:k-1}}; \theta_\triangle) - \mu)^2}{(|\mathcal{S}| - 1)!} \\
&\leq (\sigma'_{i,k})^2.
\end{aligned} \tag{32}$$

Therefore, positional Shapley value-based estimation has a smaller variance than the direct Shapley value estimation.

## H    CONFIGURATIONS

We conducted experiments on a computer with Intel(R) Xeon(R) CPU E5-2680 v4 @ 2.40GHz, RAM of 500G, and 1 GeForce RTX 2080 Ti GPU. We implemented our model with Tensor-Flow 1.15. The weights of the neural networks were randomly initialized with normal initializer. LeakyRELU was used as activation function. Adam optimizer was used for model training. The detailed model structures of SASANet are shown in Table 4.

## I    DATASET

The experiments are conducted on 3 public datasets:

Table 4: The detailed network structure of SASANet.

| Dataset | Module | Name | Value |
|---|---|---|---|
| Fraud | Feature | Embedding Dimension | 64 |
| | | Continuous Embedding | [16, 16, 64] |
| | | MLP | [128, 128, 128] |
| | Marginal | Attention Dimension | 8 |
| | | Attention Head | 4 |
| | | MLP | [128, 128, 128] |
| | Shapley | Attention Dimension | 8 |
| | | Attention Head | 8 |
| | | $\lambda_v$ | 1 |
| | | $\lambda_s$ | 1 |
| HiggsBoson | Feature | Embedding Dimension | 64 |
| | | Continuous Embedding | [128, 128, 512] |
| | | MLP | [512] * 6 + [1024] |
| | Marginal | Attention Dimension | 128 |
| | | Attention Head | 8 |
| | | MLP | [256] * 6 |
| | Shapley | Attention Dimension | 128 |
| | | Attention Head | 8 |
| | | $\lambda_v$ | 1 |
| | | $\lambda_s$ | 1 |
| Income | Feature | Embedding Dimension | 128 |
| | | Continuous Embedding | [128, 16, 128] |
| | | MLP | [256, 128, 128] |
| | Marginal | Attention Dimension | 128 |
| | | Attention Head | 16 |
| | | MLP | [256, 128, 128] |
| | Shapley | Attention Dimension | 128 |
| | | Attention Head | 16 |
| | | $\lambda_v$ | 1 |
| | | $\lambda_s$ | 0.0001 |
| Insurance | Feature | Embedding Dimension | 64 |
| | | Continuous Embedding | [16, 16] |
| | | MLP | [128, 128, 128] |
| | Marginal | Attention Dimension | 8 |
| | | Attention Head | 4 |
| | | MLP | [128, 128, 128] |
| | Shapley | Attention Dimension | 8 |
| | | Attention Head | 8 |
| | | $\lambda_v$ | 1 |
| | | $\lambda_s$ | 1 |

Table 5: Summary of the datasets.

| Dataset | Train | | | Test | | | Features |
|---|---|---|---|---|---|---|---|
| | Total | Positive | Negative | Total | Positive | Negative | |
| Income | 199,523 | 12,382 | 187,141 | 99,762 | 6,186 | 93,576 | 39 |
| Higgs Boson | 10,000,000 | 5,299,505 | 4,700,495 | 500,000 | 529,618 | 470,382 | 28 |
| Fraud | 227,845 | 380 | 227,465 | 56,962 | 112 | 56,850 | 30 |
| Insurance | 5,822 | 348 | 5,474 | 4,000 | 238 | 3762 | 85 |

**Census-Income Prediction**. This is the dataset for predicting if the income will be above 50K, given demographic and employment related features. This dataset contains anomalous census data extracted from the 1994 and 1995 Current Population Surveys conducted by the U.S. Census Bureau. We formulated the task as a binary-classification problem. The original data has been split into training and test set with a ratio of 2:1, which we have followed in our experiments.

**Higgs Boson dataset.** This is a classification problem that aims to distinguish between a signal process that produces Higgs Bosons and a background process that does not. The first 21 features

represent kinematic properties measured by the particle detectors in the accelerator. The last seven features are high-level features derived from the first 21 features to help discriminate between the two classes. In our experiments, we used the last 500,000 examples as a test set.

**Credit Card Fraud Detection** The dataset contains transactions made by credit cards in September 2013 by European cardholders. This dataset presents transactions that occurred in two days. The dataset is highly unbalanced, the positive class (frauds) account for 0.172. The dataset contains the seconds elapsed between each transaction and the first transaction in the dataset, the transaction Amount, and 28 features obtained with PCA. License: Database Contents License (DbCL) v1.0.

**Insurance Company Benchmark** This data set used in the CoIL 2000 Challenge contains information on customers of an insurance company. The data was supplied by the Dutch data mining company Sentient Machine Research and is based on a real world business problem. The data consists of 86 variables and includes product usage data and socio-demographic data.

The summary of the datasets are listed in Table 5. We have transformed each task into standard prediction tasks with structured input features. Specifically, each sample consists of an input feature vector and a class label.

## J STANDARD DEVIATION OF PERFORMANCE

Table 6: Model Performance, in which the best performance among interpretable models is in bold.

| Model | Income | | Higgs Boson | | Fraud | | Insurance | |
|---|---|---|---|---|---|---|---|---|
| | AP | AUC | AP | AUC | AP | AUC | RMSE | MAE |
| LightGBM | *0.6972* | *0.9566* | 0.8590 | 0.8459 | 0.7934 | 0.9737 | 0.2315 | *0.1052* |
| ± | 0.0001 | 0.0001 | 0.0001 | 0.0001 | 0.0122 | 0.0011 | 0.0001 | 0.0002 |
| MLP | 0.6616 | 0.9518 | *0.8877* | *0.8771* | *0.8167* | *0.9621* | *0.2310* | 0.1076 |
| ± | 0.0011 | 0.0003 | 0.0002 | 0.0001 | 0.0105 | 0.0017 | 0.0003 | 0.0008 |
| DT | 0.2514 | 0.7250 | 0.6408 | 0.6705 | 0.5639 | 0.8614 | 0.3332 | 0.1167 |
| ± | 0.0022 | 0.0018 | 0.0002 | 0.0001 | 0.0164 | 0.0073 | 0.0020 | 0.0019 |
| LR | 0.3570 | 0.8717 | 0.6835 | 0.6846 | 0.7279 | 0.9620 | - | - |
| ± | 0.0001 | 0.0001 | 0.0001 | 0.0001 | 0.0001 | 0.0001 | - | - |
| NAM | 0.6567 | 0.9506 | 0.7897 | 0.7751 | 0.7986 | 0.9590 | 0.2382 | 0.1182 |
| ± | 0.0043 | 0.0006 | 0.0003 | 0.0001 | 0.0026 | 0.0075 | 0.0018 | 0.0071 |
| SITE | 0.6415 | 0.9472 | 0.8656 | 0.8597 | 0.7912 | 0.9556 | - | - |
| ± | 0.0037 | 0.0009 | 0.0002 | 0.0011 | 0.0039 | 0.0057 | - | - |
| SENN | 0.6067 | 0.9416 | 0.7563 | 0.7556 | 0.7709 | 0.8916 | 0.2672 | 0.1313 |
| ± | 0.0013 | 0.0005 | 0.0003 | 0.0005 | 0.0004 | 0.0001 | 0.0082 | 0.0107 |
| SASANet-p | 0.6708 | 0.9525 | 0.8775 | 0.8656 | 0.8090 | 0.9667 | **0.2368** | **0.0894** |
| ± | 0.0009 | 0.0001 | 0.0008 | 0.0004 | 0.0022 | 0.0017 | 0.0010 | 0.0122 |
| SASANet-d | 0.6811 | 0.9527 | 0.8790 | 0.8675 | 0.8090 | 0.9665 | 0.2387 | 0.1037 |
| ± | 0.0009 | 0.0001 | 0.0002 | 0.0002 | 0.0020 | 0.0017 | 0.0011 | 0.0130 |
| SASANet | **0.6864** | **0.9542** | **0.8836** | **0.8721** | **0.8124** | **0.9674** | 0.2375 | 0.0901 |
| ± | 0.0002 | 0.0001 | 0.0001 | 0.0001 | 0.0004 | 0.0002 | 0.0005 | 0.0083 |

## K MEANINGFULNESS: VALUE FUNCTION EVALUATION

SASANet's value function's prediction performance given different numbers of features decides how well the modeled Shapley values reveal real-world feature-label relevance. By randomly mask each test sample to each feature set size to form separate test sets, we test the AUC performance of a trained SASANet's value function generated by the marginal contribution module on each test set. As comparison, we train a separate model for each feature size with SASANet structure (i.e., "SASANet (sep)"), which has lower fitting difficulty for only considering fixed-sized feature sets. In addition, we trained separate MLPs for each size because MLP performed the best among neural networks in Table 6. According to the results in Figure 4, SASANet is comparable to the separately trained models in terms of prediction with arbitrary-numbered features. This brings meaningful value functions and Shapley values that reflect real feature-label relevance.

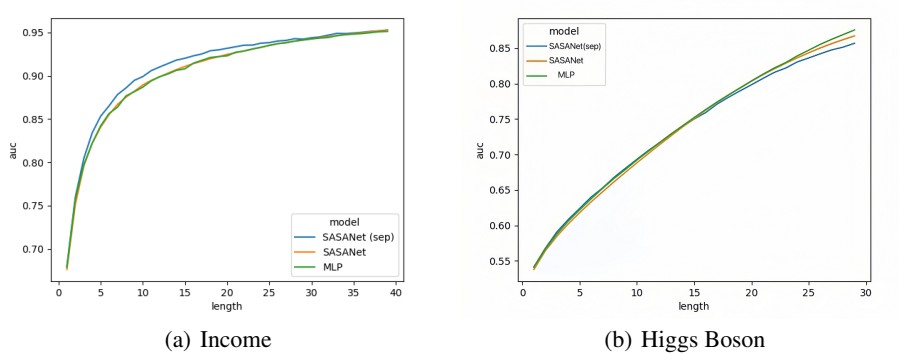

(a) Income         (b) Higgs Boson

Figure 4: AUC performance given different number of features.

## L  FEATURE ADDING EXPERIMENT

We experimented by adding features with Top-k (1-5) attribution values by different models from scratch and observe how AP and AUC increases.

Table 7: Feature-adding experiments.

| Task | Method | Top 1 | | Top 2 | | Top 3 | | Top 4 | | Top 5 | |
|---|---|---|---|---|---|---|---|---|---|---|---|
| | | AP | AUC | AP | AUC | AP | AUC | AP | AUC | AP | AUC |
| Income | SASA. | **0.448** | **0.911** | **0.511** | **0.928** | **0.582** | **0.938** | **0.603** | **0.942** | **0.617** | **0.945** |
| | KerSH. | 0.400 | 0.895 | 0.428 | 0.901 | 0.456 | 0.902 | 0.495 | 0.916 | 0.528 | 0.924 |
| | FastSH. | 0.149 | 0.679 | 0.185 | 0.528 | 0.250 | 0.756 | 0.276 | 0.770 | 0.346 | 0.840 |
| Higgs | SASA. | **0.757** | **0.756** | **0.659** | **0.659** | **0.669** | **0.685** | **0.679** | **0.701** | **0.689** | **0.714** |
| | KerSH. | 0.691 | 0.669 | 0.626 | 0.621 | 0.641 | 0.639 | 0.648 | 0.652 | 0.657 | 0.661 |
| | FastSH. | - | - | - | - | - | - | - | - | - | - |
| Fraud | SASA. | **0.519** | 0.933 | **0.747** | **0.950** | **0.791** | **0.959** | **0.809** | **0.963** | **0.812** | **0.967** |
| | KerSH. | 0.464 | **0.940** | 0.735 | 0.950 | 0.787 | 0.948 | 0.802 | 0.953 | 0.802 | 0.958 |
| | FastSH. | 0.004 | 0.673 | 0.005 | 0.737 | 0.012 | 0.824 | 0.061 | 0.866 | 0.077 | 0.875 |

## M  EXTRA VISUALIZATIONS

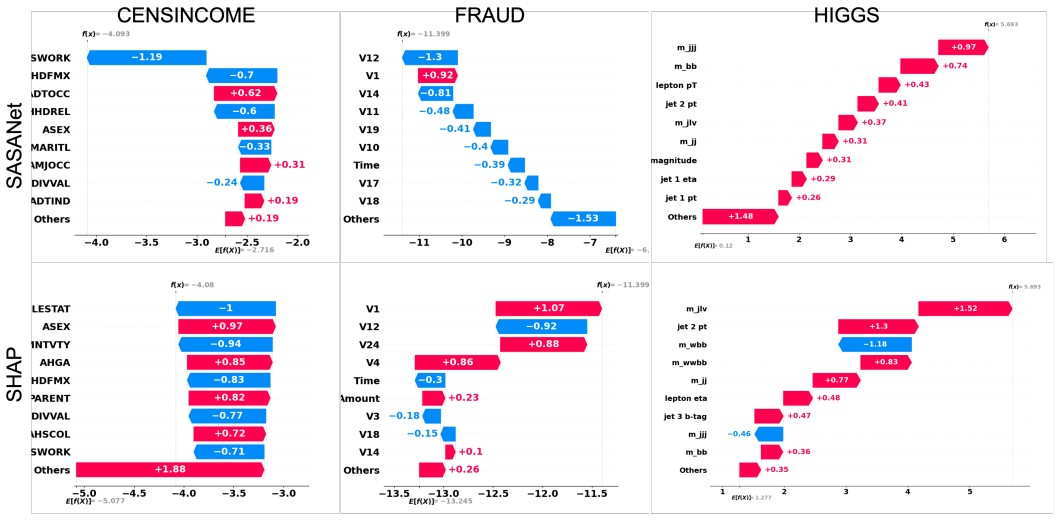

Figure 5: Additional visualizations.

