# OpenReview forum: "Towards Faithful Neural Network Intrinsic Interpretation with Shapley Additive Self-Attribution"
_ICLR.cc/2024/Conference — Submitted to ICLR 2024_

### Official Review · Reviewer_Ff8r · 2023-10-30

**Soundness:** 3 good
**Presentation:** 2 fair
**Contribution:** 2 fair
**Rating:** 5
**Confidence:** 3

**Summary:**

This paper presents a method for training an additive neural network, where the final prediction is the summation of learned contributions for each feature. The contribution for each feature is also learned on the fly, and is trained to approximate the Shapley value of each feature. On several tabular datasets, the authors show that their method is capable of attaining high performance while also learning Shapley values (i.e. contributions) for each feature which are meaningful.

**Strengths:**

### Interesting idea to have the model learn Shapley values on its own

The core concept is unique and also very interesting. Shapley values can be somewhat difficult to estimate, and it is intriguing to have them be learned from the data during training, within the framework of an interpretable additive model.

### Good mix of different datasets and benchmarks

The authors did a good job of using different tabular datasets and benchmarking across different methods that also provide interpretable features, as well as non-interpretable methods like a simple MLP.

**Weaknesses:**

### Partial marginal contribution scores might not be meaningful

I have my doubts as to how meaningful the learned marginal contribution scores from $\Delta$ can be in general. The goal is to have $\Delta$ for a single feature $i$ match its true Shapley score, but in the absence of true Shapley labels, the network is regularized to satisfy the following: 1) the summation of the $\Delta$ values over all features $i$ should be the label $y$ (Equation 3); 2) for any subset of features $\mathcal{S}$ which include $i$, different permutations of $\Delta$ on that subset should be the same value (Equation 4); and 3) partial sums of $\Delta$ values should also equal the label $y$ (Equation 5). As the authors point out, while the first two conditions are sufficient to ensure the model learns permutation-invariant contribution scores which add to the final label $y$, they alone are not sufficient to ensure that the contribution scores for each feature are meaningful (e.g. the feature contributions can be 0 for every feature except the last one). Condition 3 is meant to fix this, but it is not clear how that is done.

In particular, condition 3 (Equation 5) seems like it trains the network to make sure that the partial sums of contribution scores (i.e. for any subset of feature $\mathcal{S}$) to be equal to the label $y$ _as if_ it had the full set of features. For example, consider a 2-feature dataset where feature 1 does not matter and the true label is simply equal to feature 2. Then in Equation 5, $L_v$ on just feature 1 would try and make the importance of feature 1 match the label, but in reality the contribution of feature 1 should just be 0.

### SASANet addresses traditional Shapley challenges in a potentially inefficient way

The traditional challenges with computing Shapley values are: 1) how to tractably estimate the marginal contribution of feature $i$ when the number of subsets of features is exponential; and 2) given a subset of features, how to obtain a value/prediction when most models are not capable of handling partial inputs.

SASANet addresses these challenges by effectively training over many subsets, thereby forcing the model to be robust to fewer features, across different subsets. This might lead to significant inefficiencies in training, given the possible subsets and orderings, especially as the number of features becomes larger. The time analysis in section 4.4 is appreciated, but that seems to only include the time taken to compute the feature importances _post hoc_. How does the training time of SASANet compare to other methods?

### SASANet is only applied to small tabular datasets

In this work, SASANet is only being applied to a few small tabular datasets. Especially given the potential brittleness of the marginal contributions and the potential inefficiencies (as described above), it is possible that this method does not translate well to larger tabular datasets (e.g. gene expressions) or non-tabular datasets (e.g. images, text).

### Minor typos

- Shapley is spelled Shapely in several places
- Section 3.2, Specially → Specifically
- Typo in loss equation in 3.5

**Questions:**

In the analysis of the accuracy of Shapley values, how were ground truth values obtained for RMSE?

---

> ### Author Response · Authors · 2023-11-18
>
> Dear Reviewer Ff8r,
>
> Thank you for your time reviewing our paper and your valuable comments! Your concerns are addressed as follows.
>
> > Partial marginal contribution scores might not be meaningful
>
> The purpose of our distillation loss is not to make $\triangle$ match the Shapley value, but rather to make the learned attribution value converge to the average of our intermediate value $\triangle$ (Proposition 3.3). This will eventually make the attribution value the Shapley value of the final output (Theorem 3.4). All our losses are direct training losses instead of regularizers.
>
> The statement in condition 3, "partial sums of values should also equal the label $y$," is misunderstanding. In terms of the physical meaning of $\triangle$ and $f_c$, the objective of Equation 5 is to make the sum converges to the expectation of $y$ given these observable features (i.e., the conditional probability $p(y|x_S)$ in binary classification), which is not the label of one sample but the average in the entire dataset. This is proven in Appendix Proposition D.1. In the scenario you mentioned, "feature $x_1$ does not matter" statistically means its observation does not influence the posterior distribution of $y$. Consequently, the conditional probability $p(y|x_1)=p(y|\emptyset)$, and the contribution of $x_1$ in this case is exactly 0 because $p(y|\{x_1\})=\triangle(x_1, \emptyset) + \phi_0$ and $p(y|\emptyset)=\phi_0$.
>
> In Appendix Figure 4, we also provided a plot of the model prediction using partial observation, which demonstrates that the model can make meaningful predictions with partial observations. Our paper rigorously proves that the Shapley module converges to the desired Shapley value, which demonstrates the effectiveness of these modules.
>
>
> > SASANet addresses traditional Shapley challenges in a potentially inefficient way
>
> Modeling predictions under partially observed results is indeed a more challenging task. Our permutation-based training approach requires 2-3 times more training epochs to converge compared to directly fitting the final results. However, we believe that the benefits of obtaining accurate Shapley values, being able to model non-linear feature-label correlation, and having high robustness in predicting with missing features outweigh the additional training cost. We will try to find a space to clarify this point in our paper.
>
> > In this work, SASANet is only being applied to a few small tabular datasets. Especially given the potential brittleness of the marginal contributions and the potential inefficiencies (as described above), it is possible that this method does not translate well to larger tabular datasets (e.g. gene expressions) or non-tabular datasets (e.g. images, text).
>
> In SASANet, our main objective is to model complex non-linear correlations between semantically meaningful features and labels, which is crucial for scientific analysis and discovery. The Shapley value provides a theoretically sound approach for evaluating this correlation. However, its intricate definition makes computation highly complex and involves permutations, making it challenging to handle large-scale features effectively. In the case of high-dimensional inputs like images, text, or gene expressions, where semantics depend on spatial interactions of multiple input points, the individual points (e.g., a single pixel) may lack specific conceptual meaning to explicitly contribute to the ouput. In this case, it is more reasonable to extract concepts from raw features using backbones and attribute concept contributions to the output instead of assigning an accurate Shapley value to each input point. To achieve explainable concept-based inference tasks for CV/NLP, we simply add an attribution layer on top of the concept extraction layer. Notably, depite broad application scenarios, our paper focuses on experimenting with tabular data. Tabular data naturally have explicit concepts with clear physical meanings, thus make the pure attribution ability more explicit to observe.
>
> > Minor typos
>
> Thank you for pointing them out! We will correct it.
>
> > In the analysis of the accuracy of Shapley values, how were ground truth values obtained for RMSE?
>
> We draw different orders for each sample and directly calculate the Shapley value of the output based on its definition, i.e, sample orders and get the marginal contribution of each feature and get the average. In our experiments, we have added some engineering tricks to reduce the time complexity. For example, we pre-calculate and store the attention matrix and value among features. Then, we inrementally update the representations for each prefix subset in an order. However, it still took us days to get the ground truth results for these 1,000 samples.

---

> > ### Comment · Reviewer_Ff8r · 2023-11-20
> > **Response to comment**
> >
> > Thank you to the authors for providing additional clarifications! I have a much better understanding of the method now.
> >
> > I believe the main thing holding this work back is the limited datasets (Weakness 3 from above). Particularly because SASANet relies so heavily on permutations and partial feature sets, the concern that it cannot scale to larger and more complex datasets is very real. Particularly because the main objective is to apply SASANet to scientific discovery, it would be extremely helpful to see SASANet attain reasonable interpretability and predictive performance on a larger dataset. If the method is more limited to smaller tabular datasets where individual features need to be independently meaningful, then that is a pretty strict limitation on this method, as there are many other post hoc interpretability methods which would be much faster and more accurate on such small, simple datasets.

---

> > > ### Author Response · Authors · 2023-11-21
> > >
> > > Thank you for your reply! As we mentioned earlier, the reliance on permutations is an inherent feature of the Shapley value, regardless of whether it is used in post-hoc or self-attribution methods. Considering the widely accepted significance of the Shapley value, we believe that achieving a model that can attain Shapley self-attribution is a significant milestone. While we agree that improving efficiency can enhance the applicability of any model in various domains, and we acknowledge this as an important future direction, our work is significant within self-interpreting models that have been lacking in Shapley value-based self-attribution. It is the first to achieve a theoretically proven accurate Shapley value.
> > >
> > > The self-attribution method has undeniable advantages over post-hoc methods in terms of both accuracy and efficiency. As shown in Table 3, our method has significantly improved both efficiency and accuracy compared to post-hoc Shapley methods. With only 2-3 times the training epochs, we can instantly obtain highly accurate Shapley values for all samples when making a prediction. Therefore, we respectfully disagree that there are post-hoc substitutions for SASANet that can achieve equal accuracy in a faster way.
> > >
> > > In scientific discovery, a significant aspect involves uncovering and modeling quantitative relationships between concepts and features defined by experts, in addition to discovering from high-dimensional raw signals and observations from sensors. In our paper, we utilized two datasets: INCOME, a census dataset collected for social science research, and HIGGS, a physics dataset used to study the influence of features on the Higgs Boson procedure. These datasets represent classic and important problems. SASANet not only identifies influential features but also quantifies their complex joint interactions with labels from a statistical label expectation perspective. This provides insights into how the value of an additional concept quantitatively affects the outcome distribution.

---

### Official Review · Reviewer_2AVx · 2023-10-31

**Soundness:** 3 good
**Presentation:** 2 fair
**Contribution:** 3 good
**Rating:** 5
**Confidence:** 4

**Summary:**

This study proposes a new self-interpretable network based on Shapley values. Based on the permutation-based definition, the authors explicitly use a model to learn the marginal contribution of each input feature in each specific permutation. Then, they further learn a model to predict Shapley values of each feature based on its marginal contributions in different permutations. The task labels are used as additional supervision to ensure the performance of the network.

**Strengths:**

The network in this paper uses the multi-head attention layers to take inputs of varying lengths, which avoids problems with masking features or formulating the distribution of features.

The proposed method achieves similar performance to black-box models on tabular datasets. Besides, the authors evaluate the quality of attributions generated by the proposed network in different aspects, which demonstrate the reliability of the attribution.

**Weaknesses:**

- The proposed network is not sufficiently interpretable, because the internal modules are still black boxes. This network makes predictions while automatically providing Shapley values of each input feature. However, the modules for computing marginal contributions and Shapley values are constructed as multi-head attention layers and learned by back-propagation. Therefore, the internal computation is still unexplainable. In comparison, the SHAPNet (Wang et al., 2021) makes all the layer-wise propagation of features in the network interpretable.
- The loss function in equation (5) is confusing. It seems that the right-hand formula in equation (5) equals $L_m(x_S,y,O_S)$, because this term is independent of $(x’,y’)$. I check the proof in the appendix and I guess there is a typo: it should be $L_m(x_S,y’,O_S)$. In this way, $\sigma(f_c)$ is forced to model the distribution $p(y|x_S)$.
- The presentation of the paper needs improvement. The authors introduce the function of each module but do not provide an overview of the whole framework. Thus, when I read it the first time, it takes much time to understand the whole network. Besides, I suggest the authors clarify the models to be learned/optimized in each loss function. Otherwise, it may be confusing that whether equation (3) optimizes $\Delta(\cdot,\cdot;\theta_\Delta)$ or $f_c$.
- The scalability of the proposed method to complex tasks is questionable. First, the training and testing of the model need various permutations of input features, which leads to a huge computational cost. Second, the authors do not conduct experiments on language or image data. I wonder whether the performance of the proposed method on more complex tasks can still match the performance of black-box models, especially ResNets or transformers, which are more widely used than MLPs in applications.
- What is the computational complexity in the inference stage? Does it only use the Shapley module for inference? If yes, then what is the benefit of learning an additional marginal contribution module? If no, then the complexity in the inference stage is still $2^n$.

**Questions:**

- Why are you learning a model to predict the marginal contribution $\Delta$ instead of directly predicting $f_c(x_S)$, which is used in KernelSHAP and Frye et al., (2019)?

(Frye et al., 2019) Shapley explainability on the data manifold. In ICLR 2021.

- Typos: Section 3.5: $O_k,k$,$O_K$, $k=1$, and $O_i$ are all subscripts.

---

> ### Author Response · Authors · 2023-11-18
>
> Dear Reviewer 2AVx,
>
> Thank you for your time reviewing our paper and your valuable comments! Your concerns are addressed as follows.
>
> > Not sufficiently interpretable compared to SHAPNet
>
> SHAPNet calculates the Shapley value for each layer between hidden units, but it cannot accurately calculate the inputs' Shapley value to the final output. Our work's significance lies in being the first to realize global Shapley self-attribution, which is significant considering Shapley has been widely reconized as effective attribution metric. Indeed, attribution aims to accurately quantify the contribution of each input feature to the output, and the Shapley value is explicitly defined as the average change in output when adding a feature. It has a clear physical meaning in showing the effect of features without the need for a transparent intermediate procedure. Extensive post-hoc interpretations studies have been using the Shapley value itself as a model interpretation instead of discussing the mechanism of generating it.
>
> > The presentation of the paper needs improvement.
>
> Thank you for your suggestion and we will polish our paper accordingly!
>
> > Scalability of the proposed method to complex tasks
>
> In SASANet, our main objective is to model complex non-linear correlations between semantically meaningful features and labels, which is crucial for scientific analysis and discovery. The Shapley value provides a theoretically sound approach for evaluating this correlation. However, its intricate definition makes computation highly complex and involves permutations, making it challenging to handle large-scale features effectively. In the case of high-dimensional inputs like images, text, or gene expressions, where semantics depend on spatial interactions of multiple input points, the individual points (e.g., a single pixel) may lack specific conceptual meaning to explicitly contribute to the ouput. In this case, it is more reasonable to extract concepts from raw features using backbones and attribute concept contributions to the output instead of assigning an accurate Shapley value to each input point. To achieve explainable concept-based inference tasks for CV/NLP, we simply add an attribution layer on top of the concept extraction layer. Notably, depite broad application scenarios, our paper focuses on experimenting with tabular data. Tabular data naturally have explicit concepts with clear physical meanings, thus make the pure attribution ability more explicit to observe.
>
>
> > Computational complexity in the inference stage
>
> Only the Shapley module is used for inference. The purpose of learning marginal contribution is to converge the attribution value to the Shapley value of the final output, as shown in Theorem 3.4. Without learning marginal contribution and simply training a single Shapley module with a regularizer, we cannot obtain the Shapley value. This is because the optimization directions of the regularizer and prediction loss create a trade-off that makes it difficult to accurately converge to the constraint that the attribution value is the Shapley value of the final output.
>
> > Why learning $\triangle$ instead of $f_c(x_S)$
>
> Both approaches are theoretically feasible. Modeling $f_c(x_S)$ directly and then taking the difference to obtain $\triangle$ is theoretically equivalent to our current method and does not affect convergence of $\phi$ to the Shapley value of $f$. However, in practical implementation, modeling $\triangle$ allows the model to focus more on  the unique contribution of each added input, which aligns with Shapley's definition. On the other hand, modeling $f_c(x_S)$ requires combining the information of the entire feature set and potentially make the effect of each feature vague. Therefore, we chose to directly model $\triangle$ to simplify the operation when we design our model.
>
> > Typos about subsripts and loss equation (5):
>
> Thank you for pointing this out and we will double check the notations.

---

> > ### Comment · Reviewer_2AVx · 2023-11-21
> >
> > I thank the authors for their response. The authors have addressed most of my concerns. However, I think there is still room for improvement.
> >
> > - The scalability to complex models/tasks. I understand that the intrinsic definition of Shapley values makes it hard to handle high-dimensional input. My concern is that the architecture of the model to be explained (MLPs) in the paper is too simple and the task also seems easy. I expect experiments on more complex models to show that the proposed method *is able to* well explain and fit the performance of complex models. From another perspective, the authors can also estimate attributions of concepts in CV/NLP tasks, as they stated in the response, and then compare the explanation and performance (accuracy) of different explaining methods.
> >
> > - Given that learning $f_c(x_S)$ is also theoretically feasible, I guess learning $f_c(x_S)$ may induce less computational cost than learning $\Delta$, because the output $f_c(x_S)$ can be reused in the computation of Shapley values of different input features. In contrast, $\Delta$ is specific to each permutation of inputs. It would be better if the authors could provide some theoretical proof or empirical evidence to show that learning $\Delta$ is better.

---

> > > ### Author Response · Authors · 2023-11-21
> > >
> > > Thank you for your kind reply!
> > >
> > > - Thank you for your suggestion! In our paper, we actually did not evaluate the prediction and attribution abilities on image datasets for more rigorous evaluation. When an image dataset has explicit concept labels, making predictions based on these labels is essentially the same as modeling tabular data. However, if the image dataset does not have explicit concepts, the concepts extracted by feature extractors can have vague meanings and can only be heuristically explained as human-understandable concepts. However, many studies on model attacks suggest that there can be patterns recognized by models for prediction that are not recognizable by humans. Additionally, different feature maps can interact and be coupled to convey conceptual features. In such cases, it becomes challenging to disentangle the meaningfulness of attribution from the effect of the feature extractor. Nevertheless, the good performance of SASANet demonstrates that it can be effective when applied to extracted concepts when they can be well validated and explained. This presents another research direction that is not within the scope of our paper.
> > > - In fact, we conducted an experiment earlier to test the performance of using SASANet on top of concepts extracted from image data for prediction. The results showed that, with the same model size, switching SASANet to be on top of the image feature extractors layer resulted in comparable performance to that of a completely black-box model. However, we found this approach to be less straightforward than directly conducting experiments on tabular data. In tabular data, the performance is solely determined by the attribution layer itself without involving extra feature extractors. We did not include this information in the paper as we wanted to focus on the attribution ability for key conceptual information. If this can address your concern, we will include this information in the appendix.
> > > DATA| Model | Acc | AP | AUC |
> > > --|--|--|--|--|
> > > MNIST|CNN|0.995300|0.999159|0.999879|
> > > MNIST|SASANet|0.994600|0.999616 |0.999962|
> > > CIFAR-10| CNN|0.811400 | 0.887301 | 0.979416|
> > > CIFAR-10| SASANet|0.812300 | 0.885070 | 0.978536|
> > > - Thank you for your suggestion! $\triangle$ is not specific to a permutation, but rather to a set of prefix features (without order) and one additional feature. As a result, its input space is not significantly larger than directly modeling $f_c(x_S)$ with just one more feature. We appreciate your suggestions for providing more theoretical insights into this empirical design. We will delve into this interesting problem in the future. However, since our proof is generic and does not distinguish between these two designs, and the selection of one over the other is not the key aspect of our method, would you please exclude this as a concern while evaluating the quality of our paper?

---

### Official Review · Reviewer_JuUA · 2023-11-01

**Soundness:** 2 fair
**Presentation:** 4 excellent
**Contribution:** 4 excellent
**Rating:** 6
**Confidence:** 3

**Summary:**

The article presents a new self-interpreting approach called SASANet, having in mind to incorporate Shapley values into the additive self-attribution literature. The approach is compared to black-box approaches and empirically demonstrates their usefulness.

**Strengths:**

The approach seems theoretically well-grounded; the literature review seems thorough and complete.

**Weaknesses:**

**Major**

The only points I would like to raise concern the numerical experiments.

1 – The comparison to other self-attribution methods in Table 1 is convincing, but I think other conclusions drawn from the experiment reported in Table 1 are misleading.

1.1 - I don’t understand why comparing SASANet to interpretable approaches, more specifically to a linear regressor and a single decision tree. Those simple methods are sure to obtain somewhat deceiving performances when compared to a huge neural network, where having some black-box architecture is not a problem. Inherently interpretable predictors have advantages of their own (being fully transparent, for example) that can’t be simply quantitatively compared to others, and thus shouldn’t be compared solely in terms of performances. When it comes to interpretable machine learning, there is a huge literature on that matter, especially in an era where interpretability has been put forward for various reasons, so simple linear regressor and decision tree might just lead to underestimating the potential of interpretable approaches and overestimate the power of SASANet.

1.2 – I don’t feel like the comparison to black-box models is fair. Was the MLP of reasonable size? No detail is shared on that matter. Comparing an MLP to a richer and more complex architecture involving transformers is questionable. Also, LightGBM dates a bit (2017); the goal of those experiments is to compare SASANet to state-of-the-art approaches when it comes solely to performances (i.e. black-boxes) but I don’t feel like those baselines are sufficient.

2.1 – Table 2: I’m not sure about how to interpret this table. Why is it that having the worst performances is sought? I understand that since the top-5 relevant features were removed, but why not therefore look at the relative variation in performances?

2.2 - Fidelity experiments: It is stated in the abstract that « SASANet is shown more precise and efficient than post-hoc methods in interpreting its own predictions ». I don’t feel like this has been demonstrated in any way. The protocol for this demonstration is explained as follows: « We observed prediction performance after masking the top 1-5 features attributed by SASANet for each test sample and compared it to the outcome when using KernelSHAP, a popular post-hoc method, and FastSHAP, a recent parametric post-hoc method. The results are shown in Table 7. » And the conclusion that is drawn is the following: « SASANet’s feature masking leads to the most significant drop in performance, indicating its self-attribution is more faithful than post-hoc methods. » There is a logical gap here. Much information remains unknown: Were the features considered by the approaches correlated (to those in the top-5) in any way? Were the importance given by each approach to their top-5 approximately the same? Why not remove, for example, the features accounting for the first 20% importance? Also, Shapley values measure the difference in prediction, not the difference in performance. Therefore, a feature could have a great impact on the predictions while not affecting the performances at all (or smaller than expected). E.g. having an error of -x instead of x, thus a same squared error. Considering The protocol is simply insufficient to claim that « SASANet is shown more precise and efficient than post-hoc methods in interpreting its own predictions ».

**Minor**

1 – Typo in Theorem 3.4: « with ample permutation ».

2 – At the beginning of section 4.2: « Table 6 shows average scores from 10 tests; Appendix J lists standard deviations. »; Table 1 should be named, not Table 6. The same thing occurs later on: Table 7 is mentioned while referring to Table 2. Otherwise, Tables 1 and 2 aren’t referred to anywhere in the article.

**Questions:**

1 – When it comes to feature attribution approaches, the time is reported in Table 3, but how does the training time of SASANet compare to others?

**Details Of Ethics Concerns:**

The 9-page limit is exceeded.

---

> ### Author Response · Authors · 2023-11-18
>
> Dear Reviewer JuUA,
>
> Thank you for your time reviewing our paper and your valuable comments! Your concerns are addressed as follows.
>
> > About interpretable approaches
>
> Table 1 aims to highlight the advantages of SASANet over other self-attributing models. The two black-box models and classic interpretable models widely used in tabular data are reported to position the expressiveness of these self-attributing models. We are not trying to prove SASANet is superior to self-interpreting models under any other interpretability paradigms. Since different paradigms have different goals and formulations, we believe they cannot be compared.
>
> Under attribution paradigm, our focus is on accurately quantifying the contribution of features to the output, which has its unique significance for feature selection, debias, and discovering feature-outcome correlations. We believe that our current paper effectively demonstrates how SASANet excels other self-attribution models with its  well-grounded theoretical basis and high accuracy.
>
> > About black-box models
>
> We use these classic black-boxes commonly used for tabular data as references to demonstrate our superiority over other self-attribution methods. We have never claimed SASANet to achieve state-of-the-art black-box models' performance. It is clear that other self-attribution methods have struggled to match MLP and LightGBM, while SASANet does, showing it significantly enhances usability with guaranteed interpretability but acceptable sacrifice on expressiveness, at least nearly reaching MLP. We did not compare transformers because they are not common for tabular data. In terms of MLP size, as stated in the paper, we fine-tuned them to their best performance, since simply training an MLP has weaker supervision than SASANet's loss and more easily overfit. For instance, we find the best performance is obtained by $4 \times 64$ for CENSINCOME and $4 \times 512$ for HIGGS, which can be found in our code and we will summarize them in appendix later.
>
> > About Table 2:
>
> Feature masking is common for evaluating the accuracy of an attribution method [1]. It assesses how well the method can distinguish important features that are necessary for accurate predictions. The rationale behind is that removing an important feature for an instance make the model lacks sufficient evidence to support the right prediction. Comparing the relative performance change equals comparing the absolute performance. Since the base performance are the same for all the interperters, which is the model's original performance without removing features.
>
> [1] Alvarez Melis, et. al. "Towards robust interpretability with self-explaining neural networks." Advances in neural information processing systems 31 (2018).
>
>
> > About Fidelity experiments:
>
> In the feature-masking experiment, the correlation between a feature and the outcome is determined by the model itself, not the specific interpreter. Each interpreter's attribution is compared in terms of how good they distinguish more important features, whose removal results in larger performance reduction, remain consistent regardless of the interpreter used. Therefore, removing the same number of features and compare which interpreter makes a better selection is a fair and intuitive comparison.
> It is statistically unlikely that removing important features on which a trained model relies for accurate predictions make all samples deviate towards a prediction that maintains the original performance. In reality, once important features are removed, the model lacks sufficient evidence, and the prediction becomes more akin to guessing. A guess can be difficult to perform as good as a reasonable inference based on strong evidences.
>
> In addition to feature-masking experiment, Table 3 shows our fidelity in a different perspective, which directly indicates that we learned more accurate Shapley values than post-hoc methods.
>
>
> > Typos
>
> Thank you for pointing out the typos! We will correct them in our paper.
>
>
> > Training time of SASANet
>
> Post-hoc methods usually require no training, while incur high-cost computation for every sample. FastSHAP has a slow training process, preventing us from obtaining results on the HIGGS dataset in Table 2.
>
> Compared to regular training procedure that directly fits the final results, SASANet requires 2-3 times more epochs to converge since it learns additional information, such as Shapley values and predictions under partially observable features. However, we believe this trade-off is reasonable in practice, as it allows us to gain more information for both higher interpretability and insights into non-linear feature-label correlation, as well as an ability to robustly handle any severe missing feature problem.
>
> > Page limitation
>
> The Ethical Statements and Reproducible Statements are added according to the author guidelines, which says they are not counted towards the page limit.

---

> > ### Comment · Reviewer_JuUA · 2023-11-20
> >
> > I thank the authors for their exhaustive response.
> >
> > -Concerning Table 1 and the interpretable approaches: it is said in the response above that concerning SASANet and interpretable approaches, "since different paradigms have different goals and formulations, we believe they cannot be compared", but it is said in the article that "Black-box models outperform other compared models. Classic tree and linear methods, though interpretable, are limited by their simplicity." It mostly is that latter sentence that raised my attention, along with the presence of Linear regression and Decision tree in Table 1, when writing Weaknesses - Major - 1.1.
> >
> > -Concerning Table 1 and black-box models: I better understand the relevance of the comparison; maybe this should be highlighted more clearly in the article.
> >
> > -Concerning the fidelity experiments: I maintain that the drop in performances should be normalized by how much of feature importance have been removed. Because even though a same number of features is removed for each interpreter, this could result in significantly different amount of feature importance drop. Or how much a feature that is removed impacts the performances accordingly to the importance attribution methods have given it.

---

> > > ### Author Response · Authors · 2023-11-21
> > >
> > > Thank you for your kind reply!
> > >
> > > - We apologize for the confusion. We understand your concerns and will revise the descriptions accordingly. Perhaps this description is better: "Black-box models outperform other models in terms of prediction performance due to their complex and unrestricted black-box structure. However, high performance alone is not sufficient. Classic tree and linear methods are still popular nowadays, despite their lower prediction performance as shown in the Table. This is because they offer a significant advantage in terms of interpretability."
> > >
> > > - Thank you for your suggestion. We will revise the paper based on your suggestions.
> > >
> > > - Thank you for your suggestion. This can be an interesting new valuation method from a different perspective that we can explore. However, since our evaluation focuses on the ability to recognize important features by rank, we believe that the feature-masking experiment is appropriate. If we have understood your proposal correctly, if a feature accounts for 99% of the contribution to the prediction and the interpreter correctly recognizes it, this demonstrates the effectiveness of the interpreter. However, even if dropping this one feature causes a significant drop in performance, normalizing it would average this drop across all features with 1% contribution and may not necessarily yield a large resulting value. Therefore, this experiment is interesting but is a little bit not demonstrating the point that we want to convey.

---

### Meta-Review · Area_Chair_rAWh · 2023-12-10

**Metareview:**

The paper presents a new self-interpreting approach called SASANet based on Shapley values. The method trains an additive neural network, where the final prediction is the summation of learned contributions for each feature. The contribution for each feature is also learned on the fly, and is trained to approximate the Shapley value of each feature. The task labels are used as additional supervision to ensure the performance of the network. On several tabular datasets, the authors show that their method is capable of attaining high performance while also learning Shapley values for each feature which are meaningful. The approach is compared to black-box approaches and empirically demonstrates their usefulness.

However, the experiments can be strengthened (going beyond tabular data) to drive home the effectiveness of the method. The writing and presentation can be improved.

**Justification For Why Not Higher Score:**

While it is a promising direction to produce a model that can attain Shapley self-attribution, all reviewers agree that this work is still held back by limited datasets and experiments, and the inherent limitation of scaling to larger and more complex datasets and networks.

**Justification For Why Not Lower Score:**

N/A

---

### Decision · Program_Chairs · 2024-01-16

Reject